# Statistical Limits of Adaptive Linear Models: Low-Dimensional Estimation and Inference

**Licong Lin**
Department of Statistics
University of California, Berkeley
liconglin@berkeley.edu

**Mufang Ying**
Department of Statistics
Rutgers University - New Brunswick
my426@scarletmail.rutgers.edu

**Suvrojit Ghosh**
Department of Statistics
Rutgers University - New Brunswick
sg1565@scarletmail.rutgers.edu

**Koulik Khamaru**
Department of Statistics
Rutgers University - New Brunswick
kk1241@stat.rutgers.edu

**Cun-Hui Zhang**
Department of Statistics
Rutgers University - New Brunswick
czhang@stat.rutgers.edu

## Abstract

Estimation and inference in statistics pose significant challenges when data are collected adaptively. Even in linear models, the Ordinary Least Squares (OLS) estimator may fail to exhibit asymptotic normality for single coordinate estimation and have inflated error. This issue is highlighted by a recent minimax lower bound, which shows that the error of estimating a single coordinate can be enlarged by a multiple of $\sqrt{d}$ when data are allowed to be arbitrarily adaptive, compared with the case when they are i.i.d. Our work explores this striking difference in estimation performance between utilizing i.i.d. and adaptive data. We investigate how the degree of adaptivity in data collection impacts the performance of estimating a low-dimensional parameter component in high-dimensional linear models. We identify conditions on the data collection mechanism under which the estimation error for a low-dimensional parameter component matches its counterpart in the i.i.d. setting, up to a factor that depends on the degree of adaptivity. We show that OLS or OLS on centered data can achieve this matching error. In addition, we propose a novel estimator for single coordinate inference via solving a Two-stage Adaptive Linear Estimating equation (TALE). Under a weaker form of adaptivity in data collection, we establish an asymptotic normality property of the proposed estimator.

## 1 Introduction

Estimating a low-dimensional parameter component in a high-dimensional model is a fundamental problem in statistics and machine learning that has been widely studied in e.g., semiparametric statistics [35, 8], causal inference [16, 15] and bandit algorithms [1, 27]. When data are independently and identically distributed (i.i.d.), it is often possible to derive estimators that are asymptotically normal with a rate of convergence of $\sqrt{n}$, and that achieve the semi-parametric variance lower bound

37th Conference on Neural Information Processing Systems (NeurIPS 2023).

that is independent of the dimension. There is now a rich body of literature that studies this problem under various scenarios [5, 6, 33, 36, 4, 8, 41].

In this work we are interested in the same estimation and inference problem but under the setting where the i.i.d. data assumption fails. Specifically, we consider an adaptive collection framework where the data collected at time $i$ is allowed to be dependent on the historical data collected up to time $i - 1$. This adaptive framework incorporates datasets originated from applications in many fields, including sequential experimental design [15], bandit algorithm [27], time series modeling [7], adaptive stochastic approximation schemes [13, 24].

## 1.1 An interesting lower bound

To see the intrinsic difference between the i.i.d. and the adaptive data collection settings, we consider the canonical example of linear model $y = \boldsymbol{x}^\top \boldsymbol{\theta}^* + \varepsilon$, where the parameter $\boldsymbol{\theta}^* = (\theta_1^*, \boldsymbol{\theta}_2^*) \in \mathbb{R}^1 \times \mathbb{R}^{d-1}$, $\varepsilon \overset{iid}{\sim} \mathcal{N}(0, 1)$. Clearly, when the covariates $\{\boldsymbol{x}_i\}_{i \leq n}$ are deterministic, a straightforward calculation yields

$$\widehat{\theta}_{\mathrm{ols},1} - \theta_1^* \overset{d}{=} \mathcal{N}(0, (\mathbf{S}_n^{-1})_{11}), \quad \text{and} \quad \mathbb{E}[(\mathbf{S}_n^{-1})_{11}^{-1} \cdot (\widehat{\theta}_{\mathrm{ols},1} - \theta_1^*)^2] = 1, \tag{1}$$

where $\widehat{\theta}_{\mathrm{ols}}$ is the OLS estimator and $\mathbf{S}_n := \sum_{t=1}^n \boldsymbol{x}_i \boldsymbol{x}_i^\top$ is the sample covariance matrix.

However, somewhat surprisingly, when the covariates $\{\boldsymbol{x}_i\}_{i \leq n}$ are allowed to be collected in an *arbitrary* adaptive manner, in a recent work [20] the authors proved the following (informal) counter-intuitive minimax lower bound on the scaled-MSE (defined in Definition 2.2)

$$\min_{\widehat{\theta}} \max_{\boldsymbol{\theta}^*} \mathbb{E}[(\mathbf{S}_n^{-1})_{11}^{-1} \cdot (\widehat{\theta} - \theta_1^*)^2] \geq cd \cdot \log(n), \tag{2}$$

where the extra $d$-factor enters the estimation of a *single* coordinate. This lower bound indicates that a dimension independent single coordinate estimation is *infeasible* when the data are collected *arbitrarily* adaptively. This is undesirable especially in the high dimensional scenario where $d \to \infty$, since a $\sqrt{n}$-consistent estimation is unattainable. Motivated by the contrast between i.i.d. and adaptive data collection, we pose the following question in this work:

> *Can we bridge the gap between iid and adaptive data collection, and obtain an estimator for a low-dimensional parameter component in linear models, such that its performance depends on the degree of adaptivity?*

## 1.2 Contributions

In this work, we initiate the study of how the adaptivity of the collected data affects low-dimensional estimation in a high-dimensional linear model. We explore the previously posed question and provide an affirmative answer.

We begin by introducing a general data collection assumption, which we term $(k, d)$-*adaptivity*. Broadly speaking, $(k, d)$-adaptivity implies that the data pairs $\{(\boldsymbol{x}_i, y_i)\}_{i=1}^n \in \mathbb{R}^d \times \mathbb{R}$ are collected in a way that the first $k$ coordinates of $\boldsymbol{x}_i$ (denoted by $\boldsymbol{x}_i^{\mathrm{ad}}$) are chosen adaptively based on the historical data, while the remaining $d - k$ coordinates of $\boldsymbol{x}_i$ (denoted by $\boldsymbol{x}_i^{\mathrm{nad}}$) are i.i.d. across time $i \in [n]$.

Assume the collected data are $(k, d)-$adaptive from a linear model $y = \boldsymbol{x}^\top \boldsymbol{\theta}^* + \varepsilon$. We analyze the lower-dimensional estimation problem under the scenarios where the i.i.d. non-adaptive components $\boldsymbol{x}_i^{\mathrm{nad}}$ are either zero-mean or nonzero-mean. In the zero mean case, we show that the ordinary least squares estimator (OLS) for the first $k$-coordinate yields a scaled mean squared error (scaled-MSE) of $k \log(n)$ (Theorem 3.1). For the nonzero-mean case, a similar result is achieved using the OLS estimator on centered data (Theorem 3.2). Consequently, we find that the degree of adaptivity significantly impacts the performance of single coordinate estimation, in the sense that the scaled-MSE is inflated by a factor of $k$, where $k$ denotes the number of adaptive coordinates (see Corollary 3.3).

Although OLS for a single coordinate has a dimension independent scaled-MSE when the collected data are $(1, d)$-adaptive, it should be noted that OLS may exhibit non-normal asymptotic behavior [13, 20] when data are adaptively collected. Therefore, we propose a novel estimator by solving a Two-stage Adaptive Linear Estimating Equation (TALE). When the collected data are $(1, d)$-adaptive and the non-adaptive component is zero mean, we show that our new estimator is asymptotically normal and has a comparable scaled-MSE as the naive OLS estimator (see Theorem 3.4).

## 2 Problem set up

Consider a linear model

$$y = \boldsymbol{x}^\top \boldsymbol{\theta}^* + \varepsilon, \tag{3}$$

where the parameter $\boldsymbol{\theta}^* \in \mathbb{R}^d$, and $\varepsilon$ is a zero mean noise variable. Given access to a data set $\{(\boldsymbol{x}_i, y_i)\}_{i \leq n}$ from the model (3), we are interested in the estimation and inference problem of a low-dimensional parameter component $\boldsymbol{\theta}_{\mathrm{ad}}^* \in \mathbb{R}^k$, where $\boldsymbol{\theta}^* = (\boldsymbol{\theta}_{\mathrm{ad}}^{*\top}, \boldsymbol{\theta}_{\mathrm{nad}}^{*\top})^\top$.

In this paper, we are interested in adaptive data collection regime. Concretely, we assume that the data are collected adaptively in the following way

**Definition 2.1 ($(k, d)$-adaptivity)** *The collected samples $\{(\boldsymbol{x}_i, y_i)\}_{i \leq n}$ forms a filtration $\{\mathcal{F}\}_{i=0}^\infty$ with $\mathcal{F}_0 = \emptyset$ and $\mathcal{F}_i = \sigma(\boldsymbol{x}_1, y_1, \ldots, \boldsymbol{x}_i, y_i)$. Let $P$ be an unknown distribution on $\mathbb{R}^{d-k}$. We assume that at each stage, $i \geq 1$*

- *The adaptive component $\boldsymbol{x}_i^{\mathrm{ad}} = \boldsymbol{x}_{i,1:k}$ is collected from some unknown distribution that could depend on $\mathcal{F}_{i-1}$.*

- *The non-adaptive component $\boldsymbol{x}_i^{\mathrm{nad}} = \boldsymbol{x}_{i,k+1:d}$ is a sample from $P$ and independent of $(\boldsymbol{x}_i^{\mathrm{ad}}, \mathcal{F}_{i-1})$.*

When $k = 0$, Definition 2.1 reduces to an i.i.d. data collection strategy; when $k = d$, it corresponds to the case where the data are allowed to be collected arbitrarily adaptively. Consequently, $(k, d)$-adaptivity connects two extreme scenarios, and the degree of adaptivity increases as $k$ increases.

**Example 2.1 (Treatment assignment)** *As a concrete example, consider the problem of treatment assignment to patients. At round $i$, we observe the health profile of the patient $i$, which we denote by $\boldsymbol{x}_i \in \mathbb{R}^{d-1}$. Our job to assign a treatment $A_i \in \{0, 1\}$ based on the patient's health profile $\boldsymbol{x}_i$ and also our prior knowledge of effectiveness of the treatments. It is natural to capture our prior knowledge using $\mathcal{F}_i = \sigma(A_1, \boldsymbol{x}_1, y_1, \ldots, A_{i-1}, \boldsymbol{x}_{i-1}, y_{i-1})$ — the sigma field generated by previous data-points. As already pointed out in (2), in the adaptive regime the estimator error for treatment effect scales as $\sqrt{d/n}$ ; in words, we have to pay for a dimension factor $\sqrt{d}$ even if we are only interested in estimating a one-dimensional component. While for our treatment assignment example, the dimension $d - 1$ of the covariate vector $\boldsymbol{x}_i$ is large in practice, it is natural to assume that the treatment assignment is dependent on $k - 1 \ll d - 1$, a few (unknown) components. Under this assumption, it is easy to see that this treatment assignment problem is $(k, d)$-adaptive. We show that the treatment effect can be estimated at a rate $\sqrt{k/n} \ll \sqrt{d/n}$.*

### 2.1 Statistical limits

Before we discuss how to obtain estimators for a low-dimensional parameter component of $\theta^\star$, we establish some baselines by recalling existing lower bounds. Throughout this section, we assume the noise $\epsilon_i \overset{iid}{\sim} \mathcal{N}(0, \sigma^2)$. We start with defining the metric for comparison.

**Definition 2.2 (scaled mean squared error (scaled-MSE))** *Given a subset $\mathcal{I} \subseteq [d]$. We define the scaled-MSE of an estimator $\widehat{\boldsymbol{\theta}}_\mathcal{I}$ for $\boldsymbol{\theta}_\mathcal{I}^* \in \mathbb{R}^{|\mathcal{I}|}$ to be $\mathbb{E}[(\widehat{\boldsymbol{\theta}}_\mathcal{I} - \widehat{\boldsymbol{\theta}}_\mathcal{I})^\top [(\mathbf{S}_n^{-1})_{\mathcal{I}\mathcal{I}}]^{-1} (\widehat{\boldsymbol{\theta}}_\mathcal{I} - \widehat{\boldsymbol{\theta}}_\mathcal{I})]$, where $\mathbf{S}_n = \sum_{i=1}^n \boldsymbol{x}_i \boldsymbol{x}_i^\top$ is the sample Gram matrix.*

Roughly speaking, when the covariates $\boldsymbol{x}_i$ are all fixed, the scaled-MSE compares the performance of $\widehat{\boldsymbol{\theta}}_\mathcal{I}$ against the estimator with minimal variance (OLS). Moreover, we have the following result:

**Proposition 2.2 (A simplified version of Theorem 2 in Khamaru et al. [20])**

(a). *Given a set $\mathcal{I} \subseteq [d]$. Suppose the data $\{(\boldsymbol{x}_i, y_i)\}_{i=1^n}$ are i.i.d. ($(0, d)$-adaptive) from model (3). Then the scaled-MSE satisfies*

$$\inf_{\widehat{\boldsymbol{\theta}}} \sup_{\boldsymbol{\theta}^* \in \mathbb{R}^d} \mathbb{E} \left\| \widehat{\boldsymbol{\theta}}_\mathcal{I} - \boldsymbol{\theta}_\mathcal{I}^* \right\|_{[(\mathbf{S}_n^{-1})_{\mathcal{I}\mathcal{I}}]^{-1}}^2 \geq \sigma^2 |\mathcal{I}|. \tag{4}$$

*Furthermore, the equality holds when choosing $\widehat{\boldsymbol{\theta}}_\mathcal{I}$ to be the OLS estimator for $\boldsymbol{\theta}_\mathcal{I}^*$.*

*(b). Suppose the data points $\{(\boldsymbol{x}_i, y_i)\}_{i=1}^n$ are allowed to be arbitrarily adaptive $((d, d)$-adaptive). For any $(n, d)$ with $d \geq 2$ and $n \geq c \cdot d^3$, and any non-empty set $\mathcal{I} \in [d]$, there exists a data collection algorithm such that*

$$\inf_{\widehat{\boldsymbol{\theta}}} \sup_{\boldsymbol{\theta}^* \in \mathbb{R}^d} \mathbb{E} \left\| \widehat{\boldsymbol{\theta}}_{\mathcal{I}} - \boldsymbol{\theta}_{\mathcal{I}}^* \right\|_{[(\mathbf{S}_n^{-1})_{\mathcal{I}\mathcal{I}}]^{-1}}^2 \geq c' \cdot d\sigma^2 \log(n), \tag{5}$$

*where $c, c' > 0$ are some universal constants.*

Proposition 2.2 exhibits the striking difference between two extreme data collection mechanisms. While the scaled-MSE scales as $O(|\mathcal{I}|)$ when data are i.i.d., the scaled-MSE for even a single coordinate (e.g., setting $\mathcal{I} = \{1\}$) can be of the order $O(d)$ if the data are allowed to be collected arbitrarily adaptively.

Let $\mathcal{I}^c = [d] \setminus \mathcal{I}$. By the matrix inverse formula, we have

$$[(\mathbf{S}_n^{-1})_{\mathcal{I}\mathcal{I}}]^{-1} = (\mathbf{S}_n)_{\mathcal{I}\mathcal{I}} - (\mathbf{S}_n)_{\mathcal{I}\mathcal{I}^c}[(\mathbf{S}_n^{-1})_{\mathcal{I}^c\mathcal{I}^c}]^{-1}(\mathbf{S}_n)_{\mathcal{I}^c\mathcal{I}} = \mathbf{X}_{\mathcal{I}}^\top (\mathbf{I}_n - \mathbf{P}_{\mathbf{X}_{\mathcal{I}^c}})\mathbf{X}_{\mathcal{I}},$$

where $\mathbf{P}_{\mathbf{X}_{\mathcal{I}^c}}$ denotes the projection onto the column space of $\mathbf{X}_{\mathcal{I}^c} \in \mathbb{R}^{n \times |\mathcal{I}|}$. In this work, we are often interested in the special cases where $\mathcal{I} = [k]$ (or $\{\ell\}$ for $\ell \in [k]$), which denote (a single coordinate of) the adaptive component.

## 2.2 Related work

**Adaptive linear model**   In the early works by Lai et al. [24, 23], the authors studied regression models when the data are adaptively collected. They established the asymptotic normality of OLS under a stability assumption on the covariate matrix. However, the stability assumption might be violated under various setting, including data collected from online bandit algorithms such as UCB [27, 2, 31, 34, 42], forecasting and autoregressive models [14, 38, 24]. Recent works [13, 20] addressed this issue and proposed debiasing estimators with inferential guarantee in linear models with fixed dimension. While allowing for arbitrarily adaptively collected data, their results impose an additional $\sqrt{d}$ factor in the error bound for single coordinate estimation, limiting their applicability in linear models with increasing dimensions [19, 28].

**Parameter estimation in bandit algorithms**   Though with the primary goal being achieving a low regret, the problem of parameter estimation under adaptive data are also studied when designing online bandit algorithms [1, 26, 29, 27]. Many online bandit algorithms are built based on the estimation or construction of adaptive confidence sets for the reward function [2, 9, 17], which can be viewed as finite sample estimation and inference of the unknown parameter of a model. Most related to our paper, in linear bandits, the works [1, 26] derived non-asymptotic upper bound on scaled-MSE of OLS for the whole parameter vector, as well as for a single coordinate. However, the upper bound on scaled-MSE for estimating a single coordinate is inflated by a factor of $d$ compared with the i.i.d. case, as suggested by the lower bound in [20].

**Inference using adaptively collected data**   The problem of estimation and inference using adaptively collected data has also been studied under other settings. The work by Hadad et al. [15] and Zhan et al. [40] proposed a weighted augmented inverse propensity weighted (AIPW, [32]) estimator for treatment effect estimation that is asymptotic normal. Zhang et al. [43] analyzed a weighted $M$-estimator for contextual bandit problems. Lin et al. [30] proposed a weighted $Z$-estimator for statistical inference in semi-parametric models. While investigating more intricate models, these works are built on the strong assumption that the adaptive data collection mechanism is known. In contrast, our $(k, d)$-adaptivity assumption allows the adaptive component to be collected in an arbitrary adaptive way.

**Semi-parametric statistics**   A central problem in semi-parametric statistics is to derive $\sqrt{n}$-consistent and asymptotic normal estimators of a low-dimensional parameter component in high-dimensional or semi-parametric models [5, 6, 33, 36, 4, 8, 41]. Most works in this literature assume i.i.d. data collection and aim to obtain estimators that achieve the optimal asymptotic variance (i.e., semi-parametric efficient [35, 16]). On the other hand, our work focuses on a complementary perspective, with the goal of understanding how the data collection assumption affects the statistical limit of low-dimensional parameter estimation.

## 2.3 Notations

In the paper, we use the bold font to denote vectors and matrices (e.g., $\boldsymbol{x}, \mathbf{x}, \mathbf{X}, \boldsymbol{\theta}, \boldsymbol{\varepsilon}$), and the regular font to denote scalars (e.g., $x, \theta, \varepsilon$). Given data $\{(\boldsymbol{x}_i, y_i)\}_{i=1}^n$ from the linear model (3) that are $(k, d)$-adaptive, we use $\boldsymbol{x}_i^{\mathrm{ad}} \in \mathbb{R}^k, \boldsymbol{x}_i^{\mathrm{nad}} \in \mathbb{R}^{d-k}$ to denote the adaptive and non-adaptive covariates. We also write $\boldsymbol{\theta}^* = (\boldsymbol{\theta}_{\mathrm{ad}}^{*\top}, \boldsymbol{\theta}_{\mathrm{nad}}^{*\top})^\top$ to denote the components that correspond to the adaptive and non-adaptive covariates. Let $\mathbf{X} = (\boldsymbol{x}_1^\top, \ldots, \boldsymbol{x}_n^\top)^\top \in \mathbb{R}^{n \times d}$ be the covariate matrix, with $\mathbf{X}_{\mathrm{ad}}$ (or $\mathbf{X}_{\mathrm{nad}}$) representing the submatrices consisting of the adaptive (or non-adaptive) columns. We use $\mathbf{x}_j$ to denote the $j$-th column of the covariate matrix.

For a matrix $\mathbf{M}$ with $n$ rows, let $\mathbf{M}_{-j}$ be the matrix obtained by deleting the $j$-th column of $\mathbf{M}$. We define the *projection operator* $\mathbf{P_M} := \mathbf{M}(\mathbf{M}^\top \mathbf{M})^{-1} \mathbf{M}^\top$ and the (columnwise) centered matrix $\widetilde{\mathbf{M}} := (\mathbf{I}_n - \mathbf{P_{1_n}})\mathbf{M}$, where $\mathbf{I} \in \mathbb{R}^{n \times n}$ is the identity matrix and $\mathbf{1}_n \in \mathbb{R}^n$ is the all-one vector. For a symmetric $\mathbf{M} \succeq 0$, we define $\|\boldsymbol{x}\|_{\mathbf{M}} := \sqrt{\boldsymbol{x}^\top \mathbf{M} \boldsymbol{x}}$. Lastly, we use $c, c', c'' > 0$ to denote universal constants and $C, C', C'' > 0$ to denote constants that may depend on the problem specific parameters but not on $k, d, n$. We allow the values of the constants to vary from place to place.

# 3 Main results

This section is devoted to our main results on low-dimensional estimation and inference. In Section 3.1 and 3.2 we discuss the problem of estimating a low-dimensional component of $\theta^\star$, and Section 3.3 is devoted to inference of low-dimensional components.

## 3.1 Low-dimensional estimation

Suppose the collected data $\{(\boldsymbol{x}_i, y_i)\}_{i=1}^n$ are $(k, d)$-adaptive. In this section, we are interested in estimating the adaptive parameter component $\boldsymbol{\theta}_{\mathrm{ad}}^* \in \mathbb{R}^k$.

In addition to $(k, d)$-adaptivity, we introduce the following assumptions on the collected data $\{(\boldsymbol{x}_i, y_i)\}_{i=1}^n$.

**Assumption A**

(A1) There exists a constant $\mathrm{U_x} > 0$ such that

$$1 \le \sigma_{\min}(\mathbf{X}_{\mathrm{ad}}^\top \mathbf{X}_{\mathrm{ad}}) \le \sigma_{\max}(\mathbf{X}_{\mathrm{ad}}^\top \mathbf{X}_{\mathrm{ad}}) \le n\mathrm{U_x}.$$

(A2) The non-adaptive components $\{\boldsymbol{x}_i^{\mathrm{nad}}\}_{i=1}^n$ are i.i.d. sub-Gaussian vectors with parameter $\nu > 0$, that is, for any unit direction $\boldsymbol{u} \in \mathbb{S}^{d-1}$,

$$\mathbb{E}[\exp\{\lambda \langle \boldsymbol{u}, \boldsymbol{x}_i^{\mathrm{nad}} - \mathbb{E}[\boldsymbol{x}_i^{\mathrm{nad}}] \rangle\}] \le e^{\lambda^2 \nu^2 / 2} \qquad \forall \lambda \in \mathbb{R}.$$

(A3) There exist some constants $0 \le \sigma_{\min} \le \sigma_{\max}$ such that the covariance matrix of the non-adaptive component $\boldsymbol{\Sigma} := \mathrm{Cov}[\boldsymbol{x}_i^{\mathrm{nad}}]$ satisfies,

$$0 < \sigma_{\min} \le \sigma_{\min}(\boldsymbol{\Sigma}) \le \sigma_{\max}(\boldsymbol{\Sigma}) \le \sigma_{\max}.$$

(A4) Conditioned on $(\boldsymbol{x}_i, \mathcal{F}_{i-1})$, the noise variable $\varepsilon_i$ in (3) is zero mean sub-Gaussian with parameter $v > 0$, i.e.,

$$\mathbb{E}[\varepsilon_i | \boldsymbol{x}_i^{\mathrm{ad}}, \mathcal{F}_{i-1}] = 0, \text{ and } \mathbb{E}[e^{\lambda \varepsilon_i} | \boldsymbol{x}_i^{\mathrm{ad}}, \mathcal{F}_{i-1}] \le e^{\lambda^2 v^2 / 2} \qquad \forall \lambda \in \mathbb{R},$$

and has conditional variance $\sigma^2 = \mathbb{E}[\varepsilon_i^2 | \boldsymbol{x}_i, \mathcal{F}_{i-1}]$ for all $i \in [n]$.

Let us clarify the meaning of the above assumptions. Assumption (A1) is the regularity assumption on the adaptive component. Roughly speaking, we allow the adaptive component to be *arbitrarily adaptive* as long as $\mathbf{X}_{\mathrm{ad}}^\top \mathbf{X}_{\mathrm{ad}}$ is not close to be singular and $\boldsymbol{x}_i^{\mathrm{ad}}$ has bounded $\ell_2$−norm. This is weaker than the assumptions made in [15, 43, 30], which assume that the conditional distribution of $\mathbf{X}_{\mathrm{ad}}$ is known. Assumption (A2), (A3) on the non-adaptive component, assume its distribution is non-singular and light-tailed. Assumption (A4) is a standard assumption that characterizes the tail behavior of the zero-mean noise variable. We remark that the equal conditional variance assumption in Assumption (A4) is mainly required in Theorem 3.4, while it is sufficient to assume $\sigma^2$ being a uniform upper bound of the conditional variance in Theorem 3.1 and 3.2.

### 3.1.1 Warm up: zero-mean non-adaptive component

We start with discussing a special case where the non-adaptive component $x_i^{\mathrm{nad}}$ is zero-mean. In this case, we prove that the Ordinary Least Squares (OLS) estimator on $(\mathbf{X}, \boldsymbol{y})$ for $\boldsymbol{\theta}_{\mathrm{ad}}^*$ is near-optimal; see the discussion in Section 2.1. Denote the OLS estimator by $\widehat{\boldsymbol{\theta}} = (\widehat{\boldsymbol{\theta}}_{\mathrm{ad}}^\top, \widehat{\boldsymbol{\theta}}_{\mathrm{nad}}^\top)^\top$. Throughout, we assume that the sample size $n$ and dimension $d$ satisfies the relation

$$\frac{n}{\log^2(n/\delta)} \geq Cd^2, \tag{6}$$

where $C$ is an independent of $(n, d)$ but may depend on other problem specific parameters. With this set up, our first theorem states

**Theorem 3.1** *Given data points $\{(\boldsymbol{x}_i, y_i)\}_{i=1}^n$ from a $(k, d)$-adaptive model, and tolerance level $\delta \in (0, 1/2)$. Let, assumption (A1)–(A4) and the bound (6) in force, and the non-adaptive component $x_i^{\mathrm{nad}}$ is drawn from a zero-mean distribution. Then, we have*

$$\|\widehat{\boldsymbol{\theta}}_{\mathrm{ad}} - \boldsymbol{\theta}_{\mathrm{ad}}^*\|_{\mathbf{X}_{\mathrm{ad}}^\top(\mathbf{I}_n - \mathbf{P}_{\mathbf{X}_{\mathrm{nad}}})\mathbf{X}_{\mathrm{ad}}}^2 \leq C' \log(n \det(\mathbf{X}_{\mathrm{ad}}^\top \mathbf{X}_{\mathrm{ad}})/\delta) \tag{7a}$$

$$\leq C'' k \log(n/\delta). \tag{7b}$$

*with probability at least $1 - \delta$.*

See Appendix A.3 for a detailed proof. A few comments regarding Theorem 3.1 are in order. One might integrate both sides of the last bound to get a bound on the scaled-MSE. Comparing the bound (7b) with the lower bound from Proposition 2.2, we see this bound is tight in a minimax sense, up to some logarithmic factors.

It is now worthwhile to compare this bound with the existing best upper bounds in the literature. Invoking the concentration bounds from [26, Lemma 16] one have that

$$\|\widehat{\boldsymbol{\theta}}_{\mathrm{ad}} - \boldsymbol{\theta}_{\mathrm{ad}}^*\|_{\mathbf{X}_{\mathrm{ad}}^\top(\mathbf{I}_n - \mathbf{P}_{\mathbf{X}_{\mathrm{nad}}})\mathbf{X}_{\mathrm{ad}}}^2 \leq \|\widehat{\boldsymbol{\theta}}_{\mathrm{ad}} - \boldsymbol{\theta}_{\mathrm{ad}}^*\|_{\mathbf{X}_{\mathrm{ad}}^\top \mathbf{X}_{\mathrm{ad}}}^2 \leq c \cdot d \log(n/\delta) \tag{8}$$

One might argue that the first inequality is loose as we only want to estimate a low-dimensional component $\boldsymbol{\theta}_{\mathrm{ad}}^* \in \mathbb{R}^k$. However, invoking the lower bound from Proposition 2.2, we see that the bound (8) is the best you can hope for if we do not utilize the $(k, d)$-adaptivity structure present in the data. See also the scaled-MSE bound for a single coordinate estimation in [26, Theorem 8] which also has a dimension dependence in the scaled-MSE bound.

### 3.1.2 Nonzero-mean non-adaptive component

In practice, the assumption that the non-adaptive covariates are drawn i.i.d. from a distribution $P$ with *zero mean* is unsatisfactory. One would like to have a similar result where the distribution $P$ has an *unknown* non-zero mean.

---

**Algorithm 1** Centered OLS for $k$ adaptive coordinates $(\mathbf{X}, \boldsymbol{y})$

---

1: $\hat{\boldsymbol{\mu}}_{\mathrm{ad}} \leftarrow \frac{\mathbf{X}_{\mathrm{ad}}^\top \mathbf{1}_n}{n}$, $\hat{\boldsymbol{\mu}}_{\mathrm{nad}} \leftarrow \frac{\mathbf{X}_{\mathrm{nad}}^\top \mathbf{1}_n}{n}$
2: $\widetilde{\mathbf{X}}_{\mathrm{ad}} = \mathbf{X}_{\mathrm{ad}} - \mathbf{1}_n \hat{\boldsymbol{\mu}}_{\mathrm{ad}}^\top$, $\widetilde{\mathbf{X}}_{\mathrm{nad}} = \mathbf{X}_{\mathrm{nad}} - \mathbf{1}_n \hat{\boldsymbol{\mu}}_{\mathrm{nad}}^\top$
3: Run OLS on centered response vector $\boldsymbol{y} - \bar{y} \cdot \mathbf{1}_n$ and centered covariate matrix $\widetilde{\mathbf{X}} = (\widetilde{\mathbf{X}}_{\mathrm{ad}}, \widetilde{\mathbf{X}}_{\mathrm{nad}}) \in \mathbb{R}^{n \times d}$; obtain the estimator $\widetilde{\boldsymbol{\theta}} = (\widetilde{\boldsymbol{\theta}}_{\mathrm{ad}}^\top, \widetilde{\boldsymbol{\theta}}_{\mathrm{nad}}^\top)^\top$.

---

Before we state our estimator for the nonzero-mean case, it is helpful to understand the proof intuition of Theorem 3.1. A simple expansion yields

$$\widehat{\boldsymbol{\theta}}_{\mathrm{ad}} - \boldsymbol{\theta}_{\mathrm{ad}}^* = (\mathbf{X}_{\mathrm{ad}}^\top \mathbf{X}_{\mathrm{ad}} - \mathbf{X}_{\mathrm{ad}}^\top \mathbf{P}_{\mathbf{X}_{\mathrm{nad}}} \mathbf{X}_{\mathrm{ad}})^{-1}(\mathbf{X}_{\mathrm{ad}}^\top \boldsymbol{\varepsilon} - \mathbf{X}_{\mathrm{ad}}^\top \mathbf{P}_{\mathbf{X}_{\mathrm{nad}}} \boldsymbol{\varepsilon})$$

$$\approx (\mathbf{X}_{\mathrm{ad}}^\top \mathbf{X}_{\mathrm{ad}})^{-1} \mathbf{X}_{\mathrm{ad}}^\top \boldsymbol{\varepsilon} + \text{smaller order terms}$$

We show that the interaction term $\mathbf{X}_{\mathrm{ad}}^\top \mathbf{P}_{\mathbf{X}_{\mathrm{nad}}}$ is small compared to the other terms under $(k, d)$-adaptivity and *zero-mean* property of $\mathbf{X}_{\mathrm{nad}}$. In particular, under *zero-mean* property, each entry of

the matrix $\mathbf{X}_{\mathrm{ad}}^{\top}\mathbf{X}_{\mathrm{nad}}$ is a martingale difference sequence, and can be controlled via concentration inequalities [1]. This martingale property is *not true* when the columns of $\mathbf{X}_{\mathrm{nad}}$ have a nonzero mean.

As a remedy, we consider the mean-centered linear model:

$$\mathbf{y} - \bar{y} \cdot \mathbf{1}_n = \widetilde{\mathbf{X}}_{\mathrm{ad}}^{\top}\boldsymbol{\theta}_{\mathrm{ad}}^* + \widetilde{\mathbf{X}}_{\mathrm{nad}}^{\top}\boldsymbol{\theta}_{\mathrm{nad}}^* + (\epsilon - \bar{\epsilon} \cdot \mathbf{1}_n) \tag{9}$$

where $\widetilde{\mathbf{X}}_{\mathrm{ad}} = \mathbf{X}_{\mathrm{ad}} - \frac{\mathbf{1}_n\mathbf{1}_n^{\top}}{n}\mathbf{X}_{\mathrm{ad}}$, and $\widetilde{\mathbf{X}}_{\mathrm{nad}} = \mathbf{X}_{\mathrm{nad}} - \frac{\mathbf{1}_n\mathbf{1}_n^{\top}}{n}\mathbf{X}_{\mathrm{nad}}$ are centered version of the matrices $\mathbf{X}_{\mathrm{ad}}$ and $\mathbf{X}_{\mathrm{nad}}$, respectively. The centering in (9) ensures that $\mathbf{X}_{\mathrm{nad}}$ is *approximately* zero-mean, but unfortunately, it breaks the martingale structure present in the data. For instance, the elements of $\widetilde{\mathbf{X}}_{\mathrm{ad}}^{\top}\widetilde{\mathbf{X}}_{\mathrm{nad}}$ are not a sum of martingale difference sequence because we have subtracted the column mean from each entry. Nonetheless, it turns out that subtracting the sample mean, while breaks the martingale difference structure, does not break it in an adversarial way, and we can still control the entries of $\widetilde{\mathbf{X}}_{\mathrm{ad}}^{\top}\widetilde{\mathbf{X}}_{\mathrm{nad}}$. See Lemma A.1 part (b) for one of the key ingredient in the proof. We point out that this finding is not new. Results of this form are well understood in various forms in sequential prediction literature, albeit in a different context. Such results can be found in earlier works of Lai, Wei and Robbins [21, 22, 24] and also in the later works by several authors [11, 12, 10, 1] and the references therein.

Our following Theorem 3.2 ensures that the intuition developed in this section so far is useful to characterize the performance of the solution obtained from the centered OLS.

**Theorem 3.2** *Given data points* $\{(\boldsymbol{x}_i, y_i)\}_{i=1}^n$ *from a* $(k, d)$-*adaptive model, and tolerance level* $\delta \in (0, 1/2)$. *Let, assumption (A1)–(A4) and the bound* (6) *be in force. Then,* $\widetilde{\boldsymbol{\theta}}_{\mathrm{ad}}$ *obtained from Algorithm 1, satisfies*

$$\|\widetilde{\boldsymbol{\theta}}_{\mathrm{ad}} - \boldsymbol{\theta}_{\mathrm{ad}}^*\|^2_{\widetilde{\mathbf{X}}_{\mathrm{ad}}^{\top}(\mathbf{I}_n - \mathbf{P}_{\widetilde{\mathbf{X}}_{\mathrm{nad}}})\widetilde{\mathbf{X}}_{\mathrm{ad}}} \leq C' \log(n \det(\widetilde{\mathbf{X}}_{\mathrm{ad}}^{\top}\widetilde{\mathbf{X}}_{\mathrm{ad}})/\delta) \tag{10}$$

$$\leq C'' k \log(n/\delta).$$

*with probability at least* $1 - \delta$.

See Appendix A.4 for a proof. Note that the variance of $\widetilde{\boldsymbol{\theta}}_{\mathrm{ad}}$ is given by $\widetilde{\mathbf{X}}_{\mathrm{ad}}^{\top}(\mathbf{I}_n - \mathbf{P}_{\widetilde{\mathbf{X}}_{\mathrm{nad}}})\widetilde{\mathbf{X}}_{\mathrm{ad}}$. The covariance matrix is the same as $\mathbf{X}_{\mathrm{ad}}^{\top}(\mathbf{I}_n - \mathbf{P}_{\mathbf{X}_{\mathrm{nad}}})\mathbf{X}_{\mathrm{ad}}$ when the all one vector $\mathbf{1}_n$ belongs to the column space of $\widetilde{\mathbf{X}}_{\mathrm{nad}}$.

### 3.2 Single coordinate estimation: Application to treatment assignment

Let us now come back to Example 2.1 that we started. Let, at every round $i$, the treatment assignment $A_i$ depends on $k - 1$ (unknown) coordinate of the covariates $\boldsymbol{x}_i \in \mathbb{R}^{d-1}$. We assume that the covariates $\boldsymbol{x}_i's$ are drawn i.i.d. from some unknown distribution $\mathcal{P}$. Assuming the response is related to the treatment and covariates via a linear model, it is not hard to see that this problem satisfies a $(k, d)$-adaptivity property. The following corollary provides a bound on the estimation error of estimating the (homogeneous) treatment effect.

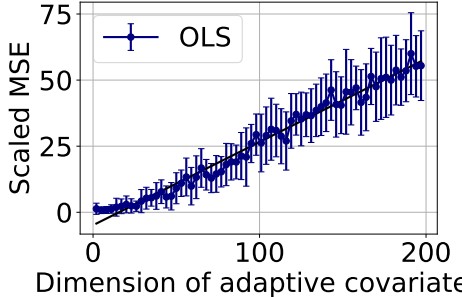 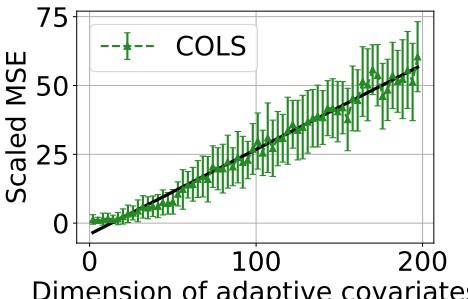

Figure 1: The plot depicts the empirical relation between the scaled MSE of the OLS and centered OLS estimate from Algorithm 1 and the number of adaptive covariates $(k)$ for a carefully constructed problem. See Section B.1 for simulation details.

**Corollary 3.3** *Suppose the assumptions from Theorem 3.2 are in force, and $\ell \in [k]$ be an index corresponding to one of the adaptive coordinates. Then, the $\ell^{th}$ coordinate of the the centered OLS estimator from Algorithm 1 satisfies*

$$|\widetilde{\theta}_{\mathrm{ad},\ell} - \theta^*_{\mathrm{ad},\ell}| \leq \frac{\sqrt{C \log(n \det(\widetilde{\mathbf{X}}^\top_{\mathrm{ad}} \widetilde{\mathbf{X}}_{\mathrm{ad}})/\delta)}}{\sqrt{\widetilde{\mathbf{x}}^\top_\ell (\mathbf{I}_n - \mathbf{P}_{\widetilde{\mathbf{X}}_{-\ell}})\widetilde{\mathbf{x}}_\ell}} \leq \frac{C\sqrt{k \log(n/\delta)}}{\sqrt{\widetilde{\mathbf{x}}^\top_\ell (\mathbf{I}_n - \mathbf{P}_{\widetilde{\mathbf{X}}_{-\ell}})\widetilde{\mathbf{x}}_\ell}}.$$

*The bounds above hold with probability at least $1 - \delta$, and $\widetilde{\mathbf{X}}_{-\ell}$ denote the matrix obtained by removing the $\ell^{th}$ column from $\widetilde{\mathbf{X}}$.*

See Appendix A.5 for a proof of this Corollary. We verify the result of Corollary 3.3 via simulations as shown in Figure 1. From the figure we see that the scaled MSE increases linearly as the number of adaptive covariates increases, matching with our theoretical predictions. See Appendix B for more details about the simulation.

### 3.3 Inference with one adaptive arm

In this section, we provide a method for constructing valid confidence intervals for $\theta^*_{\mathrm{ad}}$. To simplify the problem, we restrict our attention to the case of $(1, d)$-adaptivity and $\mathbb{E}[\boldsymbol{x}^{\mathrm{nad}}_i] = \mathbf{0}$.

**A two-stage estimator**

Our goal is to derive an asymptotically normal estimator for a target parameter in presence of a nuisance component. We call our estimator a "Two-stage-adaptive-linear-estimating-equation" based estimator, or `TALE-estimator` for short. We start with a prior estimate $\widehat{\boldsymbol{\theta}}^{\mathrm{Pr}}_{\mathrm{nad}}$ of $\theta^*_{\mathrm{nad}}$, and define our estimate $\widehat{\theta}_{\mathrm{TALE}}$ for $\theta^*_{\mathrm{ad}}$ as a solution of this

$$\texttt{TALE-estimator:} \qquad \sum_{i=1}^n w_i(y_i - x^{\mathrm{ad}}_i \cdot \widehat{\theta}_{\mathrm{TALE}} - \boldsymbol{x}^{\mathrm{nad}\top}_i \widehat{\boldsymbol{\theta}}^{\mathrm{Pr}}_{\mathrm{nad}}) = 0. \qquad (11)$$

Recall that $\widehat{\theta}_{\mathrm{TALE}}$ is a scalar and the equation has a unique solution as long as $\sum_{1 \leq i \leq n} w^{\mathrm{ad}}_i x_i \neq 0$.

The weights $\{w_i\}_{i \leq n}$ in equation (11) are a set of predictable random scalars (i.e. $w_i \in \sigma(\boldsymbol{x}^{\mathrm{ad}}_i, \mathcal{F}_{i-1})$). Specifically, we start with $s_0 > 0$ and $s_0 \in \mathcal{F}_0$, and define

$$w_i = \frac{f(s_i/s_0)x^{\mathrm{ad}}_i}{\sqrt{s_0}} \qquad \text{where} \qquad s_i = s_0 + \sum_{t \leq i}(x^{\mathrm{ad}}_t)^2 \quad \text{and} \qquad (12\mathrm{a})$$

$$f(x) = \frac{1}{\sqrt{x(\log e^2 x)(\log \log e^2 x)^2}} \qquad \text{for} \ \ x > 1. \qquad (12\mathrm{b})$$

Let us first gain some intuitions on why TALE works. By rewriting equation (11), we have

$$\sum_{i=1}^n w_i x^{\mathrm{ad}}_i \left(\widehat{\theta}_{\mathrm{TALE}} - \theta^*_{\mathrm{ad}}\right) = \underbrace{\sum_{i=1}^n w_i \epsilon_i}_{v_n} + \underbrace{\sum_{i=1}^n w_i x^{\mathrm{nad}\top}_i (\boldsymbol{\theta}^*_{\mathrm{nad}} - \widehat{\boldsymbol{\theta}}^{\mathrm{Pr}}_{\mathrm{nad}})}_{b_n}. \qquad (13)$$

Following the proof in [39], we have $v_n \xrightarrow{d} \mathcal{N}(0, \sigma^2)$. Besides, one can show that with a proper choice of prior estimator $\widehat{\boldsymbol{\theta}}^{\mathrm{Pr}}_{\mathrm{nad}}$, the bias term $b_n$ converges to zero in probability as $n$ goes to infinity. It is important to note that [39] considers the linear regression model where the number of covariates is fixed, and the sample size goes to infinity. In this work, however, we are interested in a setting where the number of covariates can grow with the number of samples. Therefore, our approach, `TALE-estimator`, has distinctions with the ALEE estimator proposed in [39]. The above intuition is formalized in the following theorem. Below, we use the shorthand $\widehat{\boldsymbol{\theta}}^{\mathrm{OLS}}_{\mathrm{nad}}$ to denote the coordinates of the least squares estimate of $\widehat{\boldsymbol{\theta}}^{\mathrm{OLS}}$ corresponding to the *non-adaptive* components.

**Theorem 3.4** *Suppose $1/s_0 + s_0/s_n = o_p(1)$, $n/(\log^2(n) \cdot d^2) \to \infty$, and assumptions (A1)-(A4) are in force. Then, the estimate $\widehat{\theta}_{TALE}$, obtained using weights from (12a) and $\widehat{\boldsymbol{\theta}}_{\mathrm{nad}}^{Pr} = \widehat{\boldsymbol{\theta}}_{\mathrm{nad}}^{OLS}$, satisfies*

$$\frac{1}{\widehat{\sigma}\sqrt{\sum_{1 \le i \le n} w_i^2}} \left( \sum_{1 \le i \le n} w_i x_i^{\mathrm{ad}} \right) \cdot \left( \widehat{\theta}_{TALE} - \theta_{\mathrm{ad}}^* \right) \xrightarrow{d} \mathcal{N}(0, 1),$$

*where $\widehat{\sigma}$ is any consistent estimate of $\sigma$. Moreover, the asymptotic variance $\widehat{\theta}_{TALE}$ is optimal up to logarithmic-factors.*

See Appendix A.6 for a proof of this theorem. The assumption $1/s_0 + s_0/s_n = o_p(1)$ in the theorem essentially requires $s_0$ grows to infinity in a rate slower than $s_n$. Therefore, in order to construct valid confidence intervals for $\widehat{\theta}_{\mathrm{TALE}}$, one has to grasp some prior knowledge about the lower bound of $s_n$. In our experiments in Section 4 we set $s_0 = \log \log(n)$. Finally, it is also worth mentioning that one can apply martingale concentration inequalities (e.g. [1]) to control the terms $b_n$ and $v_n$ in equation (13), which in turn yields the finite sample bounds for $\widehat{\theta}_{\mathrm{TALE}}$ estimator. Finally, a consistent estimator of $\sigma$ can be found using [24, Lemma 3].

## 4 Numerical experiments

In this section, we investigate the performance of TALE empirically, and compare it with the ordinary least squares (OLS) estimator, W-decorrelation proposed by Deshpande et al. [13], and the non-asymptotic confidence intervals derived from Theorem 8 in Lattimore et al. [26]. Our simulation set up entails the motivating Example 2.1 of treatment assignment. In our experiments, at stage $i$, the treatments $A_i \in \{0, 1\}$ are assigned on the sign of $\widehat{\theta}_1^{(i)}$, where $\widehat{\boldsymbol{\theta}}^{(i)} = (\widehat{\theta}_1^{(i)}, \widehat{\theta}_2^{(i)}, \dots, \widehat{\theta}_d^{(i)})$ is the least square estimate based on all data up to the time point $i-1$; here, the first coordinate of $\widehat{\theta}_1^{(i)}$ is associated with treatment assignment. The detailed data generation mechanism can be found in Appendix. From Figure 2 (top) we see that both TALE and W-decorrelation have valid empirical coverage (i.e., they are close to or above the baseline), while the nonasymptotic confidence intervals are overall conservative and the OLS is downwardly biased. In addition, TALE has confidence intervals that are shorter than those of W-decorrelation, which indicates a better estimation performance. Similar observations occur in the high-dimensional model in Figure 2 (bottom), where we find that both the OLS estimator and W-decorrelation are downwardly biased while TALE has valid coverage.

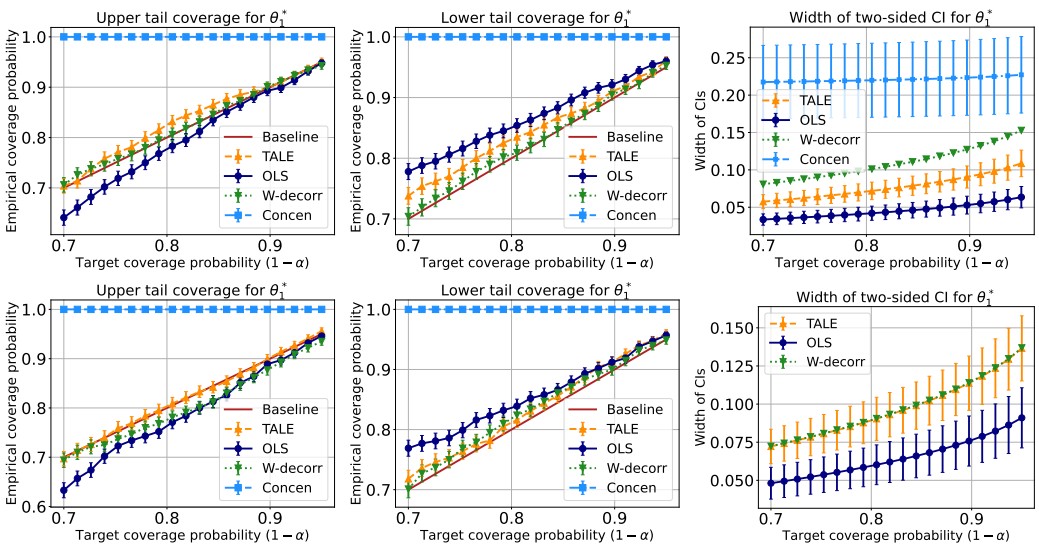

Figure 2: Empirical coverage probability and the width of confidence intervals versus target coverage probability $1 - \alpha$ for TALE, the OLS estimator, non-asymptotic concentration inequalities, and W-decorrelation. We select the noise level $\sigma = 0.3$. Top: $n = 1000, d = 10$. Bottom: $n = 500, d = 50$. At bottom right we do not display the result for concentration since the CIs are too wide. We run the simulation 1000 times and display the $\pm 1$ standard deviation.

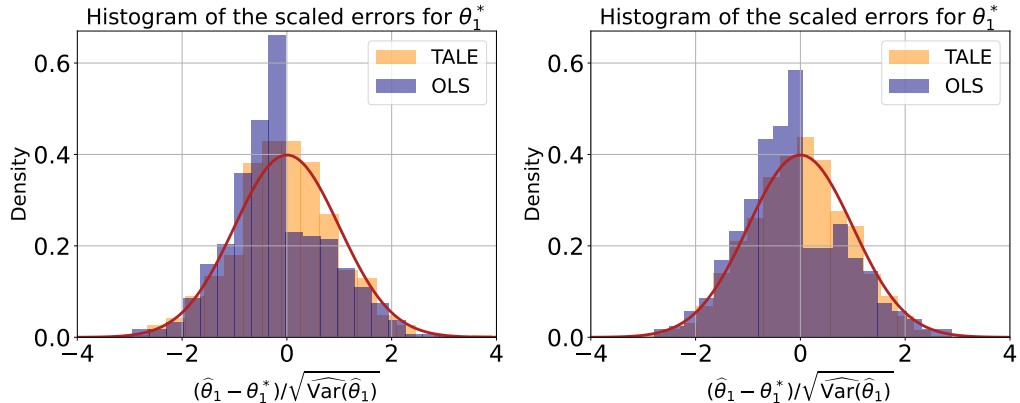

Figure 3: Histograms of the scaled errors for TALE and OLS. Left: $n = 1000, d = 10$. Right: $n = 500, d = 50$. We choose the noise level $\sigma = 0.3$ and repeat the simulation 1000 times. Observe that the distribution of the OLS estimator is much more different than standard normal, and it exhibits a downwards bias [31], while TALE is in good accordance with a standard normal distribution.

## 5 Discussion

In this paper, we investigate the statistical limits of estimating a low-dimensional component in a high dimensional adaptive linear model. We start by recalling a recent lower bound [20], which states that we need to pay for the underlying dimension $d$ even if we want to estimate a low (one)-dimensional component. Our main result is to show that in order to estimate a low-dimensional component, we need to pay only for the degree of adaptivity $k$, which can potentially be much smaller than the underlying dimension $d$. Additionally, we propose a two-stage estimator for the one-dimensional target component, which is asymptotically normal. Finally, we demonstrate the effectiveness of this two-stage estimator via numerical simulations. For the future work, there are several avenues for further exploration that can contribute to a more comprehensive understanding of adaptive regression models. First of all, it would be interesting to generalize the $(k, d)-$adaptivity for the case when the number of adaptive components may vary between samples. It is also interesting to investigate if the current assumptions can be relaxed or not. For statistical inference part, it would be interesting to extend the TALE estimator to the case when the number of adaptive components is great than one.

## Acknowledgments

This work was partially supported by the National Science Foundation Grants DMS-2311304, CCF-1934924, DMS-2052949 and DMS-2210850.

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

# Appendix

## Table of Contents

## A  Proofs

Throughout the proof, we use $c, c', c'' > 0$ to denote constants universal constants. We use $C, C', C'' > 0$ to denote constants that may depend on the problem specific parameters. Concretely, we use them to denote constants that only depends polynomially on $(1, \sigma_{\max}, 1/\sigma_{\min}, v, \nu, U_x)$. We allows the values of the constants to vary from place to place.

### A.1  Auxiliary lemmas

Before stating our main theorems, we list some useful lemmas, which can be of independent interest. All the proofs of lemmas can be found in Appendix A.

**Lemma A.1** *Given $n \geq d \geq 1$. Let $\{(\mathbf{a}_i, b_i)\}_{i=1}^n$ be a sequence of pairs such that $\mathbf{a}_i \in \mathbb{R}^d$ are $\mathcal{F}_{i-1}$-measurable and $b_i \in \mathbb{R}$ are $\mathcal{F}_i$-measurable w.r.t. some filtration $\{\mathcal{F}_i\}_{i=1}^n$. Assume in addition that $b_i$ are zero-mean sub-Gaussian random variables with parameter $\sigma$ conditioned on $\mathcal{F}_{i-1}$, i.e.,*

$$\mathbb{E}[b_i | \mathcal{F}_{i-1}] = 0, \text{ and } \mathbb{E}[e^{\lambda b_i} | \mathcal{F}_{i-1}] \leq e^{\sigma^2 \lambda^2 / 2}, \text{ for all } \lambda \in \mathbb{R}.$$

*Let $\mathbf{A} = [\mathbf{a}_1, \ldots, \mathbf{a}_n]^\top$ and $\mathbf{b} = [b_1, \ldots, b_n]^\top$. Suppose that $1 \leq \sigma_{\min}(\mathbf{A}^\top \mathbf{A}) \leq \sigma_{\max}(\mathbf{A}^\top \mathbf{A}) \leq nB$ for some constant $B > 0$.*

*(a). (A simplified version of Theorem 1 in Abbasi et al. [1].) With probability over $1 - \delta$*

$$\|\mathbf{P}_\mathbf{A} \mathbf{b}\|_2^2 = \mathbf{b}^\top \mathbf{P}_\mathbf{A} \mathbf{b} \leq c\sigma^2 \log(\det(\mathbf{A}^\top \mathbf{A})/\delta) \leq c\sigma^2 d \log(nB/\delta)$$

*for some universal constant $c > 0$.*

*(b). Let $\widetilde{\mathbf{A}} := \mathbf{A} - \mathbf{P}_{\mathbf{1}_n} \mathbf{A}$ be the centered matrix, then with probability over $1 - \delta$*

$$\|\mathbf{P}_{\widetilde{\mathbf{A}}} \mathbf{b}\|_2^2 = \mathbf{b}^\top \mathbf{P}_{\widetilde{\mathbf{A}}} \mathbf{b} \leq c\sigma^2 \log(n \det(\widetilde{\mathbf{A}}^\top \widetilde{\mathbf{A}})/\delta) \leq c\sigma^2 d \log(nB/\delta)$$

*for some universal constant $c > 0$.*

In the proofs we choose $(\mathbf{A}, \mathbf{b}) = (\mathbf{X}_{\mathrm{nad}}, \boldsymbol{\varepsilon}), (\mathbf{X}_{\mathrm{ad}}, \boldsymbol{\varepsilon}), (\mathbf{X}_{\mathrm{ad}}, \mathbf{x}_j)$ for $j \in [k+1, d]$. It is readily verified that conditions in Lemma A.1 are satisfied under assumptions in Theorem 3.1 and 3.2. See the proof of this lemma in Section A.7.

**Lemma A.2** *Suppose the data set $\{(\boldsymbol{x}_i, y_i)\}_{i=1}^n$ is $(k,d)$-adaptive and satisfies Assumption (A1)–(A4). Suppose the sample size condition* (6) *is in force. Adopt the notations in Section 3.1.2. Define* $\overline{\mathbf{X}}_{\mathrm{nad}} = \mathbf{X}_{\mathrm{nad}} - \mathbb{E}[\mathbf{X}_{\mathrm{nad}}]$ *and recall that* $\widetilde{\mathbf{X}}_{\mathrm{nad}} = (\mathbf{I}_n - \mathbf{P}_{\mathbf{1}_n})\mathbf{X}_{\mathrm{nad}}$. *For the matrices* $\overline{\mathbf{X}}_{\mathrm{nad}}, \widetilde{\mathbf{X}}_{\mathrm{nad}}$, *we have the following with probability over* $1 - \delta$

$$\frac{1}{\sigma_{\min}(\overline{\mathbf{X}}_{\mathrm{nad}})^2} = \||(\overline{\mathbf{X}}_{\mathrm{nad}}^\top \overline{\mathbf{X}}_{\mathrm{nad}})^{-1}\||_{op} \leq \frac{2}{n\sigma_{\min}} \tag{14a}$$

$$\||\overline{\mathbf{X}}_{\mathrm{nad}}\||_{op} \leq \sqrt{2n\sigma_{\max}} \tag{14b}$$

$$\||\overline{\mathbf{X}}_{\mathrm{nad}} - \widetilde{\mathbf{X}}_{\mathrm{nad}}\||_{op} \leq C\sqrt{\log(n/\delta)}\sqrt{d-k} \leq \frac{1}{2}\sigma_{\min}(\overline{\mathbf{X}}_{\mathrm{nad}}), \tag{14c}$$

$$\||(\overline{\mathbf{X}}_{\mathrm{nad}}^\top \overline{\mathbf{X}}_{\mathrm{nad}})^{-1} - (\widetilde{\mathbf{X}}_{\mathrm{nad}}^\top \widetilde{\mathbf{X}}_{\mathrm{nad}})^{-1}\||_{op} \leq \frac{C\sqrt{\log(n/\delta)}\sqrt{d-k}}{n^{3/2}}. \tag{14d}$$

$$\||\mathbf{P}_{\overline{\mathbf{X}}_{\mathrm{nad}}} - \mathbf{P}_{\widetilde{\mathbf{X}}_{\mathrm{nad}}}\||_{op} \leq C\sqrt{\log(n/\delta)}\sqrt{\frac{d-k}{n}} \leq \frac{1}{4} \tag{14e}$$

$$\|(\mathbf{P}_{\overline{\mathbf{X}}_{\mathrm{nad}}} - \mathbf{P}_{\widetilde{\mathbf{X}}_{\mathrm{nad}}})\boldsymbol{\varepsilon}\|_2 \leq C \tag{14f}$$

*for some parameter-dependent constants $C > 0$. Moreover, equation* (14a), (14b) *also hold when replacing* $\overline{\mathbf{X}}_{\mathrm{nad}}$ *with the zero-mean matrix* $\mathbf{X}_{\mathrm{nad}}$ *defined in Section 3.1.1.*

See the proof in Section A.8.

**Lemma A.3** *Under assumptions in Theorem 3.1, with probability over* $1 - \delta$

$$\mathbf{X}_{\mathrm{ad}}^\top \mathbf{P}_{\mathbf{X}_{\mathrm{nad}}} \mathbf{X}_{\mathrm{ad}} \preceq \frac{C(d-k)k\log(n/\delta)}{n}\mathbf{X}_{\mathrm{ad}}^\top \mathbf{X}_{\mathrm{ad}}$$

*for some parameter-dependent constant $C > 0$.*

See the proof in Section A.9.

**Lemma A.4** *Under assumptions in Theorem 3.2, with probability over* $1 - \delta$

$$\widetilde{\mathbf{X}}_{\mathrm{ad}}^\top \mathbf{P}_{\widetilde{\mathbf{X}}_{\mathrm{nad}}} \widetilde{\mathbf{X}}_{\mathrm{ad}} \preceq \frac{C(d-k)k\log(n/\delta)}{n}\widetilde{\mathbf{X}}_{\mathrm{ad}}^\top \widetilde{\mathbf{X}}_{\mathrm{ad}}$$

*for some parameter-dependent constant $C > 0$.*

See the proof in Section A.10.

## A.2 Proof of Proposition 2.2

**Part (a).** Proposition 2.2(a) is a standard result on the minimax optimality of the OLS estimator in linear models. A proof of this result using a Bayes argument can be found in the proof of Theorem 2(a) in Khamaru et al. [20].

By properties of the OLS estimator, we have $(\widehat{\boldsymbol{\theta}}_{\mathrm{ols}} - \boldsymbol{\theta}^*) \sim \mathcal{N}(0, \sigma^2(\mathbf{S}_n)^{-1})$ conditioned on $\mathbf{X}$. Therefore $(\widehat{\boldsymbol{\theta}}_{\mathrm{ols},\mathcal{I}} - \boldsymbol{\theta}^*_{\mathcal{I}}) \sim \mathcal{N}(0, \sigma^2(\mathbf{S}_n^{-1})_{\mathcal{I}\mathcal{I}})$ conditioned on $\mathbf{X}$, where $(\mathbf{S}_n^{-1})_{\mathcal{I}\mathcal{I}}$ denotes the submatrix of $\mathbf{S}_n^{-1}$ that consists of the coordinates in $\mathcal{I}$. It follows immediately that

$$\mathbb{E}\left\|\widehat{\boldsymbol{\theta}}_{\mathcal{I}} - \boldsymbol{\theta}^*_{\mathcal{I}}\right\|_{[(\mathbf{S}_n^{-1})_{\mathcal{I}\mathcal{I}}]^{-1}}^2 = \sigma^2\operatorname{tr}(\mathbf{I}_{|\mathcal{I}|}) = \sigma^2|\mathcal{I}|$$

for any $\boldsymbol{\theta}^* \in \mathbb{R}^d$. As a result, the OLS estimator attains the minimax lower bound when the data are i.i.d.

**Part (b).** When $|\mathcal{I}| = 1$, part (b) follows immediately from Theorem 2(b) of Khamaru et al. [20] with $\mathbf{v}$ chosen to be the one-hot vector supported on $\mathcal{I}$. When $|\mathcal{I}| > 1$, w.l.o.g. assume $\mathcal{I}$ consists of the first $|\mathcal{I}|$ coordinates of $[d]$. It suffices to show the scaled-MSE for $\mathcal{I}$ is always no less than the scaled-MSE for the first coordinate.

This follows from properties of Schur complement and the projection operator,

$$
\left\|\widehat{\boldsymbol{\theta}}_{\mathcal{I}} - \boldsymbol{\theta}_{\mathcal{I}}^*\right\|_{[(\mathbf{S}_n^{-1})_{\mathcal{I}\mathcal{I}}]^{-1}}^2
$$
$$
= (\widehat{\boldsymbol{\theta}}_{\mathcal{I}} - \boldsymbol{\theta}_{\mathcal{I}}^*)^\top \mathbf{X}_{\mathcal{I}}^\top (\mathbf{I}_n - \mathbf{P}_{\mathbf{X}_{\mathcal{I}^c}}) \mathbf{X}_{\mathcal{I}} (\widehat{\boldsymbol{\theta}}_{\mathcal{I}} - \boldsymbol{\theta}_{\mathcal{I}}^*)
$$
$$
= (\widehat{\boldsymbol{\theta}}_{\mathcal{I}} - \boldsymbol{\theta}_{\mathcal{I}}^*)^\top \begin{pmatrix} 1 & \star \\ 0 & \mathbf{I}_{k-1} \end{pmatrix} \begin{pmatrix} d_1 & \mathbf{0}^\top \\ \mathbf{0} & \mathbf{D}_2 \end{pmatrix} \begin{pmatrix} 1 & 0 \\ \star & \mathbf{I}_{k-1} \end{pmatrix} (\widehat{\boldsymbol{\theta}}_{\mathcal{I}} - \boldsymbol{\theta}_{\mathcal{I}}^*)
$$
$$
= (\widehat{\theta}_1 - \theta_1^* \quad \star) \begin{pmatrix} d_1 & \mathbf{0}^\top \\ \mathbf{0} & \mathbf{D}_2 \end{pmatrix} \begin{pmatrix} \widehat{\theta}_1 - \theta_1^* \\ \star \end{pmatrix}
$$
$$
\geq d_1(\widehat{\theta}_1 - \theta_1^*)^2,
$$

where $\mathbf{P}_{\mathbf{M}}^{\perp}$ denote the projection onto the orthogonal space of the column space of $\mathbf{M}$ and

$$
d_1 = \mathbf{x}_1^\top \mathbf{P}_{\mathbf{X}_{\mathcal{I}^c}}^{\perp} \mathbf{x}_1 - \mathbf{x}_1^\top \mathbf{P}_{\mathbf{X}_{\mathcal{I}^c}}^{\perp} \mathbf{X}_{\mathcal{I},-1} (\mathbf{X}_{\mathcal{I},-1}^\top \mathbf{P}_{\mathbf{X}_{\mathcal{I}^c}}^{\perp} \mathbf{X}_{\mathcal{I},-1})^{-1} \mathbf{X}_{\mathcal{I},-1}^\top \mathbf{P}_{\mathbf{X}_{\mathcal{I}^c}}^{\perp} \mathbf{x}_1,
$$
$$
\mathbf{D}_2 = \mathbf{X}_{\mathcal{I},-1}^\top \mathbf{P}_{\mathbf{X}_{\mathcal{I}^c}}^{\perp} \mathbf{X}_{\mathcal{I},-1}.
$$

Using the properties of projection operator and linear space decomposition, it can be verified that

$$
d_1 = \mathbf{x}_1^\top \mathbf{P}_{\mathbf{X}_{-1}}^{\perp} \mathbf{x}_1 = \mathbf{x}_1^\top (\mathbf{I}_n - \mathbf{P}_{\mathbf{X}_{-1}}) \mathbf{x}_1.
$$

Therefore $\left\|\widehat{\boldsymbol{\theta}}_{\mathcal{I}} - \boldsymbol{\theta}_{\mathcal{I}}^*\right\|_{[(\mathbf{S}_n^{-1})_{\mathcal{I}\mathcal{I}}]^{-1}}^2 \geq \left\|\widehat{\theta}_1 - \theta_1^*\right\|_{[(\mathbf{S}_n^{-1})_{11}]^{-1}}^2$. This completes the proof.

## A.3 Proof of Theorem 3.1

By the definition of the OLS estimator, we have

$$
\widehat{\boldsymbol{\theta}} - \boldsymbol{\theta}^* = \begin{pmatrix} \mathbf{X}_{\mathrm{ad}}^\top \mathbf{X}_{\mathrm{ad}} & \mathbf{X}_{\mathrm{ad}}^\top \mathbf{X}_{\mathrm{nad}} \\ \mathbf{X}_{\mathrm{nad}}^\top \mathbf{X}_{\mathrm{ad}} & \mathbf{X}_{\mathrm{nad}}^\top \mathbf{X}_{\mathrm{nad}} \end{pmatrix}^{-1} \cdot \begin{pmatrix} \mathbf{X}_{\mathrm{ad}}^\top \boldsymbol{\varepsilon} \\ \mathbf{X}_{\mathrm{nad}}^\top \boldsymbol{\varepsilon} \end{pmatrix}.
$$

Applying the block matrix inverse formula, we obtain

$$
\widehat{\boldsymbol{\theta}}_{\mathrm{ad}} - \boldsymbol{\theta}_{\mathrm{ad}}^* = (\mathbf{X}_{\mathrm{ad}}^\top \mathbf{X}_{\mathrm{ad}} - \mathbf{X}_{\mathrm{ad}}^\top \mathbf{P}_{\mathbf{X}_{\mathrm{nad}}} \mathbf{X}_{\mathrm{ad}})^{-1} (\mathbf{X}_{\mathrm{ad}}^\top \boldsymbol{\varepsilon} - \mathbf{X}_{\mathrm{ad}}^\top \mathbf{P}_{\mathbf{X}_{\mathrm{nad}}} \boldsymbol{\varepsilon}).
$$

To simplify notation, we define

$$
\overline{\boldsymbol{R}}_1 := \mathbf{X}_{\mathrm{ad}}^\top \mathbf{X}_{\mathrm{ad}} - \mathbf{X}_{\mathrm{ad}}^\top \mathbf{P}_{\mathbf{X}_{\mathrm{nad}}} \mathbf{X}_{\mathrm{ad}}, \qquad \overline{\boldsymbol{R}}_2 := \mathbf{X}_{\mathrm{ad}}^\top \boldsymbol{\varepsilon} - \mathbf{X}_{\mathrm{ad}}^\top \mathbf{P}_{\mathbf{X}_{\mathrm{nad}}} \boldsymbol{\varepsilon},
$$

and let

$$
\boldsymbol{R}_1 := \mathbf{X}_{\mathrm{ad}}^\top \mathbf{X}_{\mathrm{ad}}, \qquad \boldsymbol{R}_2 := \mathbf{X}_{\mathrm{ad}}^\top \boldsymbol{\varepsilon}.
$$

Therefore

$$
(\widehat{\boldsymbol{\theta}}_{\mathrm{ad}} - \boldsymbol{\theta}_{\mathrm{ad}}^*)^\top \overline{\boldsymbol{R}}_1 (\widehat{\boldsymbol{\theta}}_{\mathrm{ad}} - \boldsymbol{\theta}_{\mathrm{ad}}^*) = \overline{\boldsymbol{R}}_2^\top \overline{\boldsymbol{R}}_1^{-1} \overline{\boldsymbol{R}}_2.
$$

We claim the following results which we prove later. With probability over $1 - \delta$

$$
\mathbf{0} \preceq \frac{1}{2}\boldsymbol{R}_1 \preceq \overline{\boldsymbol{R}}_1 \preceq \boldsymbol{R}_1, \tag{15a}
$$

$$
|\overline{\boldsymbol{R}}_2^\top \overline{\boldsymbol{R}}_1^{-1} \overline{\boldsymbol{R}}_2 - \boldsymbol{R}_2^\top \boldsymbol{R}_1^{-1} \boldsymbol{R}_2| \leq C \log(n \det(\mathbf{X}_{\mathrm{ad}}^\top \mathbf{X}_{\mathrm{ad}})/\delta), \tag{15b}
$$

$$
\boldsymbol{R}_2^\top \boldsymbol{R}_1^{-1} \boldsymbol{R}_2 \leq C \log(\det(\mathbf{X}_{\mathrm{ad}}^\top \mathbf{X}_{\mathrm{ad}})/\delta), \tag{15c}
$$

Taking these claims as given, we establish

$$
(\widehat{\boldsymbol{\theta}}_{\mathrm{ad}} - \boldsymbol{\theta}_{\mathrm{ad}}^*)^\top \overline{\boldsymbol{R}}_1 (\widehat{\boldsymbol{\theta}}_{\mathrm{ad}} - \boldsymbol{\theta}_{\mathrm{ad}}^*) = \overline{\boldsymbol{R}}_2^\top \overline{\boldsymbol{R}}_1^{-1} \overline{\boldsymbol{R}}_2 \leq 2\overline{\boldsymbol{R}}_2^\top \boldsymbol{R}_1^{-1} \overline{\boldsymbol{R}}_2, \tag{16}
$$

$$
\overline{\boldsymbol{R}}_2^\top \boldsymbol{R}_1^{-1} \overline{\boldsymbol{R}}_2 \leq |\overline{\boldsymbol{R}}_2^\top \boldsymbol{R}_1^{-1} \overline{\boldsymbol{R}}_2 - \boldsymbol{R}_2^\top \boldsymbol{R}_1^{-1} \boldsymbol{R}_2| + \boldsymbol{R}_2^\top \boldsymbol{R}_1^{-1} \boldsymbol{R}_2 \leq C \log(n \det(\mathbf{X}_{\mathrm{ad}}^\top \mathbf{X}_{\mathrm{ad}})/\delta) \tag{17}
$$

where equation (16) uses claim (15a) and equation (17) uses claim (15b), (15c). Putting the last two displays together completes the proof.

**Proof of claim** (15a) The first and third inequality follows from the definition of $\overline{\boldsymbol{R}}_1$ and $\boldsymbol{R}_1$. The second inequality follows from Lemma A.3 and noting that $\frac{C(d-k)k\log(n/\delta)}{n} < 1/2$ under the sample size assumption (6) with $C$ in (6) chosen sufficiently large.

**Proof of claim** (15b) Define $\widetilde{\mathbf{P}}_{\mathbf{A}} := (\mathbf{A}^\top\mathbf{A})^{-1/2}\mathbf{A}^\top$ for any $\mathbf{A} \in \mathbb{R}^{n\times d}$. Then we have $\|\widetilde{\mathbf{P}}_{\mathbf{A}}\mathbf{b}\|_2 = \|\mathbf{P}_{\mathbf{A}}\mathbf{b}\|_2$ for any $\mathbf{b} \in \mathbb{R}^n$. Note that

$$
|\overline{\boldsymbol{R}}_2^\top \boldsymbol{R}_1^{-1}\overline{\boldsymbol{R}}_2 - \boldsymbol{R}_2^\top \boldsymbol{R}_1^{-1}\boldsymbol{R}_2|
$$
$$
\leq |(\overline{\boldsymbol{R}}_2 - \boldsymbol{R}_2)^\top \boldsymbol{R}_1^{-1}(\overline{\boldsymbol{R}}_2 - \boldsymbol{R}_2)| + 2|(\overline{\boldsymbol{R}}_2 - \boldsymbol{R}_2)^\top \boldsymbol{R}_1^{-1}\boldsymbol{R}_2|
$$
$$
= \varepsilon^\top \mathbf{P}_{\mathbf{X}_{\mathrm{nad}}}\mathbf{P}_{\mathbf{X}_{\mathrm{ad}}}\mathbf{P}_{\mathbf{X}_{\mathrm{nad}}}\varepsilon + 2|\varepsilon^\top \mathbf{P}_{\mathbf{X}_{\mathrm{nad}}}\mathbf{P}_{\mathbf{X}_{\mathrm{ad}}}\varepsilon|
$$
$$
= \varepsilon^\top \widetilde{\mathbf{P}}_{\mathbf{X}_{\mathrm{nad}}}(\mathbf{X}_{\mathrm{nad}}^\top\mathbf{X}_{\mathrm{nad}})^{-1/2}(\mathbf{X}_{\mathrm{nad}}^\top\mathbf{P}_{\mathbf{X}_{\mathrm{ad}}}\mathbf{X}_{\mathrm{nad}})(\mathbf{X}_{\mathrm{nad}}^\top\mathbf{X}_{\mathrm{nad}})^{-1/2}\widetilde{\mathbf{P}}_{\mathbf{X}_{\mathrm{nad}}}\varepsilon
$$
$$
\quad + 2|\varepsilon^\top \widetilde{\mathbf{P}}_{\mathbf{X}_{\mathrm{nad}}}(\mathbf{X}_{\mathrm{nad}}^\top\mathbf{X}_{\mathrm{nad}})^{-1/2}\mathbf{X}_{\mathrm{nad}}^\top\mathbf{P}_{\mathbf{X}_{\mathrm{ad}}}\varepsilon|
$$
$$
\leq \frac{\|\mathbf{X}_{\mathrm{nad}}^\top\mathbf{P}_{\mathbf{X}_{\mathrm{ad}}}\mathbf{X}_{\mathrm{nad}}\|_{\mathrm{op}}}{\sigma_{\min}(\mathbf{X}_{\mathrm{nad}}^\top\mathbf{X}_{\mathrm{nad}})}\|\widetilde{\mathbf{P}}_{\mathbf{X}_{\mathrm{nad}}}\varepsilon\|_2^2 + 2\frac{\|\mathbf{X}_{\mathrm{nad}}^\top\mathbf{P}_{\mathbf{X}_{\mathrm{ad}}}\varepsilon\|_2}{\sigma_{\min}(\mathbf{X}_{\mathrm{nad}}^\top\mathbf{X}_{\mathrm{nad}})^{1/2}}\|\widetilde{\mathbf{P}}_{\mathbf{X}_{\mathrm{nad}}}\varepsilon\|_2.
$$

Applying equation (14a) and Lemma A.1 in the last display, we continue

$$
|\overline{\boldsymbol{R}}_2^\top \boldsymbol{R}_1^{-1}\overline{\boldsymbol{R}}_2 - \boldsymbol{R}_2^\top \boldsymbol{R}_1^{-1}\boldsymbol{R}_2|
$$
$$
\leq \frac{C}{n}\mathrm{tr}(\mathbf{X}_{\mathrm{nad}}^\top\mathbf{P}_{\mathbf{X}_{\mathrm{ad}}}\mathbf{X}_{\mathrm{nad}})\|\widetilde{\mathbf{P}}_{\mathbf{X}_{\mathrm{nad}}}\varepsilon\|_2^2 + \frac{C\|\mathbf{X}_{\mathrm{nad}}^\top\widetilde{\mathbf{P}}_{\mathbf{X}_{\mathrm{ad}}}\|_{\mathrm{F}}\|\widetilde{\mathbf{P}}_{\mathbf{X}_{\mathrm{ad}}}\varepsilon\|_2}{\sqrt{n}}\|\widetilde{\mathbf{P}}_{\mathbf{X}_{\mathrm{nad}}}\varepsilon\|_2
$$
$$
\leq \frac{C(d-k)}{n}\max_{k+1\leq j\leq d}(\mathbf{x}_j^\top\mathbf{P}_{\mathbf{X}_{\mathrm{ad}}}\mathbf{x}_j)\|\mathbf{P}_{\mathbf{X}_{\mathrm{nad}}}\varepsilon\|_2^2
$$
$$
\quad + \frac{C\sqrt{k}\max_{k+1\leq j\leq d}(\sqrt{\mathbf{x}_j^\top\mathbf{P}_{\mathbf{X}_{\mathrm{ad}}}\mathbf{x}_j})\|\mathbf{P}_{\mathbf{X}_{\mathrm{ad}}}\varepsilon\|_2}{\sqrt{n}}\|\mathbf{P}_{\mathbf{X}_{\mathrm{nad}}}\varepsilon\|_2
$$
$$
\leq \frac{C(d-k)^2}{n}\log(n\det(\mathbf{X}_{\mathrm{ad}}^\top\mathbf{X}_{\mathrm{ad}})/\delta)\log(n/\delta) + \frac{C\sqrt{k(d-k)}}{\sqrt{n}}\log(n\det(\mathbf{X}_{\mathrm{ad}}^\top\mathbf{X}_{\mathrm{ad}})/\delta)\log^{1/2}(n/\delta)
$$
$$
\leq \log(n\det(\mathbf{X}_{\mathrm{ad}}^\top\mathbf{X}_{\mathrm{ad}})/\delta),
$$

where the fourth line uses Lemma A.1 and equation (14a), the last line follows from the sample size assumption (6).

**Proof of claim** (15c) This is a direct consequence of Lemma A.1 since $\varepsilon_i$ are conditionally zero-mean sub-Gaussian by Assumption (A4).

## A.4 Proof of Theorem 3.2

Let $\boldsymbol{\mu}^* = (\boldsymbol{\mu}_{\mathrm{ad}}^{*\top}, \boldsymbol{\mu}_{\mathrm{nad}}^{*\top})^\top$ denote the mean vector of $\mathbb{E}[\boldsymbol{x}_i]$ and define

$$
\overline{\mathbf{X}}_{\mathrm{nad}} := \mathbf{X}_{\mathrm{nad}} - \mathbf{1}_n\boldsymbol{\mu}_{\mathrm{nad}}^{*\top}.
$$

We write $\overline{\mathbf{X}}_{\mathrm{nad}} = [\overline{\mathbf{x}}_{k+1} \quad \ldots \quad \overline{\mathbf{x}}_d]$. The proof of this theorem follows the same basic steps as the proof of Theorem 3.1.

Recall that $\widehat{\boldsymbol{\theta}}_{\mathrm{ad}} \in \mathbb{R}^k$ denotes the non-adaptive component of the centered OLS estimator. By definition and the matrix inverse formula, we have

$$
\widehat{\boldsymbol{\theta}}_{\mathrm{ad}} - \boldsymbol{\theta}_{\mathrm{ad}}^* = (\widetilde{\mathbf{X}}_{\mathrm{ad}}^\top(\mathbf{I}_n - \mathbf{P}_{\widetilde{\mathbf{X}}_{\mathrm{nad}}})\widetilde{\mathbf{X}}_{\mathrm{ad}})^{-1} \cdot \widetilde{\mathbf{X}}_{\mathrm{ad}}^\top(\mathbf{I}_n - \mathbf{P}_{\widetilde{\mathbf{X}}_{\mathrm{nad}}})\varepsilon.
$$

To simplify notation, we introduce

$$
\widetilde{\boldsymbol{R}}_3 := \widetilde{\mathbf{X}}_{\mathrm{ad}}^\top(\mathbf{I}_n - \mathbf{P}_{\widetilde{\mathbf{X}}_{\mathrm{nad}}})\widetilde{\mathbf{X}}_{\mathrm{ad}}, \qquad \widetilde{\boldsymbol{R}}_4 := \widetilde{\mathbf{X}}_{\mathrm{ad}}^\top(\mathbf{I}_n - \mathbf{P}_{\widetilde{\mathbf{X}}_{\mathrm{nad}}})\varepsilon,
$$

and

$$
\overline{\boldsymbol{R}}_3 := \widetilde{\mathbf{X}}_{\mathrm{ad}}^\top(\mathbf{I}_n - \mathbf{P}_{\overline{\mathbf{X}}_{\mathrm{nad}}})\widetilde{\mathbf{X}}_{\mathrm{ad}}, \qquad \overline{\boldsymbol{R}}_4 := \widetilde{\mathbf{X}}_{\mathrm{ad}}^\top(\mathbf{I}_n - \mathbf{P}_{\overline{\mathbf{X}}_{\mathrm{nad}}})\varepsilon,
$$
$$
\boldsymbol{R}_3 := \widetilde{\mathbf{X}}_{\mathrm{ad}}^\top\widetilde{\mathbf{X}}_{\mathrm{ad}}, \qquad\qquad\qquad \boldsymbol{R}_4 := \widetilde{\mathbf{X}}_{\mathrm{ad}}^\top\varepsilon,
$$

Consequently,

$$(\widehat{\boldsymbol{\theta}}_{\mathrm{ad}} - \boldsymbol{\theta}^*_{\mathrm{ad}})^\top \widetilde{\boldsymbol{R}}_3 (\widehat{\boldsymbol{\theta}}_{\mathrm{ad}} - \boldsymbol{\theta}^*_{\mathrm{ad}}) = \widetilde{\boldsymbol{R}}_4^\top \widetilde{\boldsymbol{R}}_3^{-1} \widetilde{\boldsymbol{R}}_4.$$

Again, we claim the following results which we prove later. With probability over $1 - \delta$

$$\mathbf{0} \preceq \frac{1}{2} \boldsymbol{R}_3 \preceq \widetilde{\boldsymbol{R}}_3 \preceq \boldsymbol{R}_3, \tag{18a}$$

$$|\widetilde{\boldsymbol{R}}_4^\top \boldsymbol{R}_3^{-1} \widetilde{\boldsymbol{R}}_4 - \boldsymbol{R}_4^\top \boldsymbol{R}_3^{-1} \boldsymbol{R}_4| \leq C \log(n \det(\widetilde{\mathbf{X}}_{\mathrm{ad}}^\top \widetilde{\mathbf{X}}_{\mathrm{ad}})/\delta), \tag{18b}$$

$$\boldsymbol{R}_4^\top \boldsymbol{R}_3^{-1} \boldsymbol{R}_4 \leq C \log(n \det(\widetilde{\mathbf{X}}_{\mathrm{ad}}^\top \widetilde{\mathbf{X}}_{\mathrm{ad}})/\delta). \tag{18c}$$

With the claims at hand, we obtain

$$(\widehat{\boldsymbol{\theta}}_{\mathrm{ad}} - \boldsymbol{\theta}^*_{\mathrm{ad}})^\top \widetilde{\boldsymbol{R}}_3 (\widehat{\boldsymbol{\theta}}_{\mathrm{ad}} - \boldsymbol{\theta}^*_{\mathrm{ad}}) = \widetilde{\boldsymbol{R}}_4^\top \widetilde{\boldsymbol{R}}_3^{-1} \widetilde{\boldsymbol{R}}_4 \leq 2\widetilde{\boldsymbol{R}}_4^\top \boldsymbol{R}_3^{-1} \widetilde{\boldsymbol{R}}_4, \tag{19}$$

$$\widetilde{\boldsymbol{R}}_4^\top \boldsymbol{R}_3^{-1} \widetilde{\boldsymbol{R}}_4 \leq |\widetilde{\boldsymbol{R}}_4^\top \boldsymbol{R}_3^{-1} \widetilde{\boldsymbol{R}}_4 - \boldsymbol{R}_4^\top \boldsymbol{R}_3^{-1} \boldsymbol{R}_4| + \boldsymbol{R}_4^\top \boldsymbol{R}_3^{-1} \boldsymbol{R}_4 \leq C \log(n \det(\widetilde{\mathbf{X}}_{\mathrm{ad}}^\top \widetilde{\mathbf{X}}_{\mathrm{ad}})/\delta) \tag{20}$$

where equation (19) uses claim (18a) and equation (20) uses claim (18b), (18c). Combining the last two displays concludes the proof.

**Proof of claim** (18a)   The first and third inequality follows from the definition of $\widetilde{\boldsymbol{R}}_3$ and $\boldsymbol{R}_3$. For the second inequality, we have

$$\begin{aligned}
\boldsymbol{R}_3 - \widetilde{\boldsymbol{R}}_3 &= (\boldsymbol{R}_3 - \overline{\boldsymbol{R}}_3) + (\overline{\boldsymbol{R}}_3 - \widetilde{\boldsymbol{R}}_3) \\
&= \widetilde{\mathbf{X}}_{\mathrm{ad}}^\top \mathbf{P}_{\overline{\mathbf{X}}_{\mathrm{nad}}} \widetilde{\mathbf{X}}_{\mathrm{ad}} + \widetilde{\mathbf{X}}_{\mathrm{ad}}^\top (\mathbf{P}_{\widetilde{\mathbf{X}}_{\mathrm{nad}}} - \mathbf{P}_{\overline{\mathbf{X}}_{\mathrm{nad}}}) \widetilde{\mathbf{X}}_{\mathrm{ad}} \\
&\preceq \frac{1}{4} \widetilde{\mathbf{X}}_{\mathrm{ad}}^\top \widetilde{\mathbf{X}}_{\mathrm{ad}} + \frac{1}{4} \widetilde{\mathbf{X}}_{\mathrm{ad}}^\top \widetilde{\mathbf{X}}_{\mathrm{ad}} = \frac{1}{2} \boldsymbol{R}_3,
\end{aligned}$$

where the last line uses Lemma A.2, A.4 and noting that $\frac{C(d-k)k \log(n/\delta)}{n} < 1/4$ under our sample size assumption (6) with $C$ in (6) chosen sufficiently large.

**Proof of claim** (18b)   Recall that we define $\widetilde{\mathbf{P}}_{\mathbf{A}} := (\mathbf{A}^\top \mathbf{A})^{-1/2} \mathbf{A}^\top$ for any $\mathbf{A} \in \mathbb{R}^{n \times d}$. Note that

$$\begin{aligned}
&|\widetilde{\boldsymbol{R}}_4^\top \boldsymbol{R}_3^{-1} \widetilde{\boldsymbol{R}}_4 - \boldsymbol{R}_4^\top \boldsymbol{R}_3^{-1} \boldsymbol{R}_4| \\
&\leq |(\widetilde{\boldsymbol{R}}_4 - \boldsymbol{R}_4)^\top \boldsymbol{R}_3^{-1} (\widetilde{\boldsymbol{R}}_4 - \boldsymbol{R}_4)| + 2|(\widetilde{\boldsymbol{R}}_4 - \boldsymbol{R}_4)^\top \boldsymbol{R}_3^{-1} \boldsymbol{R}_4| \\
&\leq 2[(\widetilde{\boldsymbol{R}}_4 - \overline{\boldsymbol{R}}_4)^\top \boldsymbol{R}_3^{-1} (\widetilde{\boldsymbol{R}}_4 - \overline{\boldsymbol{R}}_4) + (\overline{\boldsymbol{R}}_4 - \boldsymbol{R}_4)^\top \boldsymbol{R}_3^{-1} (\overline{\boldsymbol{R}}_4 - \boldsymbol{R}_4) \\
&\quad + |(\widetilde{\boldsymbol{R}}_4 - \overline{\boldsymbol{R}}_4)^\top \boldsymbol{R}_3^{-1} \boldsymbol{R}_4| + |(\overline{\boldsymbol{R}}_4 - \boldsymbol{R}_4)^\top \boldsymbol{R}_3^{-1} \boldsymbol{R}_4|] \\
&=: 2[W_1 + W_2 + W_3 + W_4],
\end{aligned}$$

where

$$\begin{aligned}
W_1 &:= \boldsymbol{\varepsilon}^\top (\mathbf{P}_{\overline{\mathbf{X}}_{\mathrm{nad}}} - \mathbf{P}_{\widetilde{\mathbf{X}}_{\mathrm{nad}}}) \mathbf{P}_{\widetilde{\mathbf{X}}_{\mathrm{ad}}} (\mathbf{P}_{\overline{\mathbf{X}}_{\mathrm{nad}}} - \mathbf{P}_{\widetilde{\mathbf{X}}_{\mathrm{nad}}}) \boldsymbol{\varepsilon} \\
W_2 &:= |\boldsymbol{\varepsilon}^\top (\mathbf{P}_{\overline{\mathbf{X}}_{\mathrm{nad}}} - \mathbf{P}_{\widetilde{\mathbf{X}}_{\mathrm{nad}}}) \mathbf{P}_{\widetilde{\mathbf{X}}_{\mathrm{ad}}} \boldsymbol{\varepsilon}| \\
W_3 &:= \boldsymbol{\varepsilon}^\top \mathbf{P}_{\overline{\mathbf{X}}_{\mathrm{nad}}} \mathbf{P}_{\widetilde{\mathbf{X}}_{\mathrm{ad}}} \mathbf{P}_{\overline{\mathbf{X}}_{\mathrm{nad}}} \boldsymbol{\varepsilon} \\
W_4 &:= |\boldsymbol{\varepsilon}^\top \mathbf{P}_{\overline{\mathbf{X}}_{\mathrm{nad}}} \mathbf{P}_{\widetilde{\mathbf{X}}_{\mathrm{ad}}} \boldsymbol{\varepsilon}|.
\end{aligned}$$

We next bound $W_i (i = 1, 2, 3, 4)$ respectively.

For $W_1$ and $W_2$, we have from Lemma A.1 and equation (14f) that

$$W_1 \leq \|(\mathbf{P}_{\overline{\mathbf{X}}_{\mathrm{nad}}} - \mathbf{P}_{\widetilde{\mathbf{X}}_{\mathrm{nad}}}) \boldsymbol{\varepsilon}\|_2^2 \leq C$$

$$W_2 \leq \|\boldsymbol{\varepsilon}^\top (\mathbf{P}_{\overline{\mathbf{X}}_{\mathrm{nad}}} - \mathbf{P}_{\widetilde{\mathbf{X}}_{\mathrm{nad}}})\|_2 \|\mathbf{P}_{\widetilde{\mathbf{X}}_{\mathrm{ad}}} \boldsymbol{\varepsilon}\|_2 \leq C \sqrt{\log(n \det(\widetilde{\mathbf{X}}_{\mathrm{ad}}^\top \widetilde{\mathbf{X}}_{\mathrm{ad}})/\delta)}.$$

For $W_3$, similar to the proof of claim (15b), we have

$$
\begin{aligned}
W_3 &= \boldsymbol{\varepsilon}^\top \widetilde{\mathbf{P}}_{\overline{\mathbf{X}}_{\mathrm{nad}}} (\overline{\mathbf{X}}_{\mathrm{nad}}^\top \overline{\mathbf{X}}_{\mathrm{nad}})^{-1/2} \overline{\mathbf{X}}_{\mathrm{nad}}^\top \mathbf{P}_{\widetilde{\mathbf{X}}_{\mathrm{ad}}} \overline{\mathbf{X}}_{\mathrm{nad}} (\overline{\mathbf{X}}_{\mathrm{nad}}^\top \overline{\mathbf{X}}_{\mathrm{nad}})^{-1/2} \widetilde{\mathbf{P}}_{\overline{\mathbf{X}}_{\mathrm{nad}}} \boldsymbol{\varepsilon} \\
&\leq \frac{(d-k) \max_{k+1\leq j\leq d} \overline{\mathbf{x}}_j^\top \mathbf{P}_{\widetilde{\mathbf{X}}_{\mathrm{ad}}} \overline{\mathbf{x}}_j}{\sigma_{\min}(\overline{\mathbf{X}}_{\mathrm{nad}}^\top \overline{\mathbf{X}}_{\mathrm{nad}})} \|\widetilde{\mathbf{P}}_{\overline{\mathbf{X}}_{\mathrm{nad}}} \boldsymbol{\varepsilon}\|_2^2 \\
&\leq \frac{C(d-k)^2}{n} \log(n \det(\widetilde{\mathbf{X}}_{\mathrm{ad}}^\top \widetilde{\mathbf{X}}_{\mathrm{ad}})/\delta) \log(n/\delta) \\
&\leq C \log(n \det(\widetilde{\mathbf{X}}_{\mathrm{ad}}^\top \widetilde{\mathbf{X}}_{\mathrm{ad}})/\delta),
\end{aligned}
$$

where third line follows from Lemma A.1 and the last line follows from the sample size assumption (6). Likewise,

$$
\begin{aligned}
W_4 &= |\boldsymbol{\varepsilon}^\top \mathbf{P}_{\overline{\mathbf{X}}_{\mathrm{nad}}} \mathbf{P}_{\widetilde{\mathbf{X}}_{\mathrm{ad}}} \boldsymbol{\varepsilon}| \\
&\leq \|\widetilde{\mathbf{P}}_{\overline{\mathbf{X}}_{\mathrm{nad}}} \boldsymbol{\varepsilon}\|_2 \|(\overline{\mathbf{X}}_{\mathrm{nad}}^\top \overline{\mathbf{X}}_{\mathrm{nad}})^{-1/2}\|_{\mathrm{op}} \|\overline{\mathbf{X}}_{\mathrm{nad}}^\top \mathbf{P}_{\widetilde{\mathbf{X}}_{\mathrm{ad}}} \boldsymbol{\varepsilon}\|_2 \\
&\leq \frac{c\sqrt{d-k}}{\sigma_{\min}(\overline{\mathbf{X}}_{\mathrm{nad}}^\top \overline{\mathbf{X}}_{\mathrm{nad}})^{1/2}} \max_{k+1\leq j\leq d} \|\overline{\mathbf{x}}_j^\top \widetilde{\mathbf{P}}_{\widetilde{\mathbf{X}}_{\mathrm{ad}}}\|_2 \|\widetilde{\mathbf{P}}_{\widetilde{\mathbf{X}}_{\mathrm{ad}}} \boldsymbol{\varepsilon}\|_2 \|\widetilde{\mathbf{P}}_{\overline{\mathbf{X}}_{\mathrm{nad}}} \boldsymbol{\varepsilon}\|_2 \\
&\leq \frac{C(d-k)}{\sqrt{n}} \log(n \det(\widetilde{\mathbf{X}}_{\mathrm{ad}}^\top \widetilde{\mathbf{X}}_{\mathrm{ad}})/\delta) \sqrt{\log(n/\delta)} \\
&\leq C \log(n \det(\widetilde{\mathbf{X}}_{\mathrm{ad}}^\top \widetilde{\mathbf{X}}_{\mathrm{ad}})/\delta).
\end{aligned}
$$

where the third inequality follows from Lemma A.1 again. Putting pieces together yields the desired result.

**Proof of claim** (18c)  This is a direct consequence of Lemma A.1.

## A.5  Proof of Corollary 3.3

W.l.o.g. assume $\ell = 1$. By Schur decomposition, we have

$$
\begin{aligned}
&(\widetilde{\boldsymbol{\theta}}_{\mathrm{ad}} - \boldsymbol{\theta}_{\mathrm{ad}}^*)^\top \widetilde{\mathbf{X}}_{\mathrm{ad}}^\top \widetilde{\mathbf{X}}_{\mathrm{ad}} (\widetilde{\boldsymbol{\theta}}_{\mathrm{ad}} - \boldsymbol{\theta}_{\mathrm{ad}}^*) \\
&= (\widetilde{\boldsymbol{\theta}}_{\mathrm{ad}} - \boldsymbol{\theta}_{\mathrm{ad}}^*)^\top \begin{pmatrix} 1 & \widetilde{\mathbf{x}}_1^\top \widetilde{\mathbf{X}}_{\mathrm{ad},-1} \left( \widetilde{\mathbf{X}}_{\mathrm{ad},-1}^\top \widetilde{\mathbf{X}}_{\mathrm{ad},-1} \right)^{-1} \\ 0 & \mathbf{I}_{k-1} \end{pmatrix} \begin{pmatrix} \widetilde{\mathbf{x}}_1^\top (\mathbf{I}_n - \mathbf{P}_{\widetilde{\mathbf{X}}_{\mathrm{ad},-1}}) \widetilde{\mathbf{x}}_1 & \mathbf{0}^\top \\ \mathbf{0} & \widetilde{\mathbf{X}}_{\mathrm{ad},-1}^\top \widetilde{\mathbf{X}}_{\mathrm{ad},-1} \end{pmatrix} \\
&\quad \cdot \begin{pmatrix} 1 & 0 \\ \left( \widetilde{\mathbf{X}}_{\mathrm{ad},-1}^\top \widetilde{\mathbf{X}}_{\mathrm{ad},-1} \right)^{-1} \widetilde{\mathbf{X}}_{\mathrm{ad},-1}^\top \widetilde{\mathbf{x}}_1 & \mathbf{I}_{k-1} \end{pmatrix} (\widetilde{\boldsymbol{\theta}}_{\mathrm{ad}} - \boldsymbol{\theta}_{\mathrm{ad}}^*) \\
&= \begin{pmatrix} \widetilde{\theta}_{\mathrm{ad},1} - \theta_{\mathrm{ad},1}^* & \star \end{pmatrix} \begin{pmatrix} \widetilde{\mathbf{x}}_1^\top (\mathbf{I}_n - \mathbf{P}_{\widetilde{\mathbf{X}}_{\mathrm{ad},-1}}) \widetilde{\mathbf{x}}_1 & \mathbf{0}^\top \\ \mathbf{0} & \widetilde{\mathbf{X}}_{\mathrm{ad},-1}^\top \widetilde{\mathbf{X}}_{\mathrm{ad},-1} \end{pmatrix} \begin{pmatrix} \widetilde{\theta}_{\mathrm{ad},1} - \theta_{\mathrm{ad},1}^* \\ \star \end{pmatrix} \\
&\geq (\widetilde{\theta}_{\mathrm{ad},1} - \theta_{\mathrm{ad},1}^*)^2 (\widetilde{\mathbf{x}}_1^\top (\mathbf{I}_n - \mathbf{P}_{\widetilde{\mathbf{X}}_{\mathrm{ad},-1}}) \widetilde{\mathbf{x}}_1).
\end{aligned}
$$

Therefore by Lemma A.4, the sample size assumption (6), and Theorem 3.2, we establish

$$
\begin{aligned}
&(\widetilde{\theta}_{\mathrm{ad},1} - \theta_{\mathrm{ad},1}^*)^2 (\widetilde{\mathbf{x}}_1^\top (\mathbf{I}_n - \mathbf{P}_{\widetilde{\mathbf{X}}_{\mathrm{ad},-1}}) \widetilde{\mathbf{x}}_1) \\
&\leq (\widetilde{\boldsymbol{\theta}}_{\mathrm{ad}} - \boldsymbol{\theta}_{\mathrm{ad}}^*)^\top \widetilde{\mathbf{X}}_{\mathrm{ad}}^\top \widetilde{\mathbf{X}}_{\mathrm{ad}} (\widetilde{\boldsymbol{\theta}}_{\mathrm{ad}} - \boldsymbol{\theta}_{\mathrm{ad}}^*) \leq C (\widetilde{\boldsymbol{\theta}}_{\mathrm{ad}} - \boldsymbol{\theta}_{\mathrm{ad}}^*)^\top \widetilde{\mathbf{X}}_{\mathrm{ad}}^\top (\mathbf{I}_n - \mathbf{P}_{\widetilde{\mathbf{X}}_{\mathrm{nad}}}) \widetilde{\mathbf{X}}_{\mathrm{ad}} (\widetilde{\boldsymbol{\theta}}_{\mathrm{ad}} - \boldsymbol{\theta}_{\mathrm{ad}}^*) \\
&\leq C \log(n \det(\widetilde{\mathbf{X}}_{\mathrm{ad}}^\top \widetilde{\mathbf{X}}_{\mathrm{ad}})/\delta) \leq Ck \log(n/\delta).
\end{aligned}
$$

Corollary 3.3 follows immediately from the fact that $\mathbf{I}_n - \mathbf{P}_{\widetilde{\mathbf{X}}_{-1}} \preceq \mathbf{I}_n - \mathbf{P}_{\widetilde{\mathbf{X}}_{\mathrm{ad},-1}}$ since $\widetilde{\mathbf{X}}_{\mathrm{ad},-1}$ consists of some columns of $\widetilde{\mathbf{X}}_{-1}$.

## A.6 Proof of Theorem 3.4

We start with the following decomposition

$$\sum_{i=1}^{n} w_i x_i^{\mathrm{ad}} \left( \widehat{\theta}_{\mathrm{TALE}} - \theta_{\mathrm{ad}}^* \right) = \underbrace{\sum_{i=1}^{n} w_i \epsilon_i}_{v_n} + \underbrace{\sum_{i=1}^{n} w_i \boldsymbol{x}_i^{\mathrm{nad}\top} (\boldsymbol{\theta}_{\mathrm{nad}}^* - \widehat{\boldsymbol{\theta}}_{\mathrm{nad}}^{\mathrm{Pr}})}_{b_n}.$$

To prove the theorem, it suffices to show

$$v_n \xrightarrow{d} \mathcal{N}(0, \alpha \cdot \sigma^2) \qquad \text{and} \qquad b_n \xrightarrow{p} 0,$$

where $\xrightarrow{d}$ stands for convergence in distribution and $\alpha$ is a constant that is specified in equation (21).

**Proof of $v_n \xrightarrow{d} \mathcal{N}(0, \alpha \cdot \sigma^2)$:** The proof of this part directly follows from [39, Theorem 3.1]. For completeness, we provide a proof here. Note that function $f$ is a positive decreasing function and satisfies properties

$$\int_1^{\infty} f(x)dx = \infty \qquad \text{and} \qquad \int_1^{\infty} f^2(x)dx = \alpha. \tag{21}$$

Furthermore, it can be shown that

$$\max_{1 \leq i \leq n} f^2\left(\frac{s_i}{s_0}\right)\frac{(x_i^{\mathrm{ad}})^2}{s_0} = o_p(1) \quad \text{and} \quad \max_{1 \leq i \leq n} \left(1 - \frac{f(s_i/s_0)}{f(s_{i-1}/s_0)}\right) = o_p(1). \tag{22}$$

Next we compute

$$\sum_{i=1}^{n} w_i^2 = \sum_{i=1}^{n} f^2(s_i/s_0)\frac{(x_i^{\mathrm{ad}})^2}{s_0} = \int_1^{s_n/s_0} f^2(x)dx \cdot \frac{\sum_{t \leq n} f^2(s_i/s_0)(x_i^{\mathrm{ad}})^2/s_0}{\int_1^{s_n/s_0} f^2(x)dx}$$

$$\overset{(i)}{=} \int_1^{s_n/s_0} f^2(x)dx \cdot \left(1 + \frac{\sum_{i \leq n}(\frac{f^2(s_i/s_0)}{f^2(\xi_i/s_0)} - 1)f^2(\xi_i/s_0)(x_i^{\mathrm{ad}})^2/s_0}{\sum_{i \leq n} f^2(\xi_i/s_0)(x_i^{\mathrm{ad}})^2/s_0}\right).$$

In equation $(i)$, we consider the mean value theorem where $\int_{s_{i-1}/s_0}^{s_i/s_0} f^2(x)dx = f^2(\xi_i/s_0)(x_i^{\mathrm{ad}})^2/s_0$. Consequently, we have

$$\frac{\sum_{i \leq n} |\frac{f^2(s_i/s_0)}{f^2(\xi_i/s_0)} - 1|f^2(\xi_i/s_0)(x_i^{\mathrm{ad}})^2/s_0}{\sum_{i \leq n} f(\xi_i/s_0)(x_i^{\mathrm{ad}})^2/s_0} \leq \frac{\sum_{i \leq n} |\frac{f^2(s_i/s_0)}{f^2(s_{i-1}/s_0)} - 1|f^2(\xi_i/s_0)(x_i^{\mathrm{ad}})^2/s_0}{\sum_{i \leq n} f^2(\xi_i/s_0)(x_i^{\mathrm{ad}})^2/s_0}$$

$$\leq \max_{i \leq n}\left(1 - \frac{f^2(s_i/s_0)}{f^2(s_{i-1}/s_0)}\right) = o_p(1).$$

We conclude that

$$\sum_{i=1}^{n} w_i^2 = (1 + o_p(1))\int_1^{s_n/s_0} f^2(x)dx = \alpha + o_p(1) \tag{23}$$

By noticing $\max_{1 \leq i \leq n} w_i^2 = \max_{1 \leq i \leq n} f^2(s_i/s_0)(x_i^{\mathrm{ad}})^2/s_0^2 = o_p(1)$, we conclude from martingale central limit theorem that

$$\sum_{i=1}^{n} w_i \varepsilon_i \xrightarrow{d} \mathcal{N}(0, \alpha \cdot \sigma^2).$$

Moreover, applying Slutsky's theorem yields

$$\frac{1}{\widehat{\sigma}(\sum_{1 \leq i \leq n} w_i^2)^{1/2}}\left(\sum_{i=1}^{n} w_i \varepsilon_i\right) \xrightarrow{d} \mathcal{N}(0, 1). \tag{24}$$

**Proof of $b_n \xrightarrow{p} 0$:** To simplify notations, let $\boldsymbol{w} = (w_1, \ldots, w_n)^\top$. Without loss of generality, we consider the first column of the design matrix is collected adaptively. By the definition of $b_n$, we observe that

$$
\left| \sum_{i=1}^n w_i \boldsymbol{x}_i^{\mathrm{nad}\top} (\widehat{\boldsymbol{\theta}}_{\mathrm{nad}}^{\mathrm{Pr}} - \boldsymbol{\theta}_{\mathrm{nad}}^*) \right| \leq \| \sum_{i=1}^n w_i \boldsymbol{x}_i^{\mathrm{nad}} \|_2 \cdot \| \widehat{\boldsymbol{\theta}}_{\mathrm{nad}}^{\mathrm{Pr}} - \boldsymbol{\theta}_{\mathrm{nad}}^* \|_2
$$

$$
= \underbrace{\sqrt{\sum_{i=2}^d (\boldsymbol{w}^\top \mathbf{x}_i^{\mathrm{nad}})^2}}_{\triangleq b_{n,1}} \cdot \underbrace{\| \widehat{\boldsymbol{\theta}}_{\mathrm{nad}}^{\mathrm{Pr}} - \boldsymbol{\theta}_{\mathrm{nad}}^* \|_2}_{\triangleq b_{n,2}} \tag{25}
$$

**Analysis of $b_{n,1}$:** By the construction of the weights $\{w_i\}_{1 \leq i \leq n}$, we have

$$
\|\boldsymbol{w}\|_2^2 \leq \int_1^\infty f^2(x) dx = \alpha. \tag{26}
$$

Applying Lemma A.1 with $\mathbf{A} = \boldsymbol{w}$ and $b = \mathbf{x}_i^{\mathrm{nad}}$, we conclude that with probability at least $1 - \delta$,

$$
(\boldsymbol{w}^\top \mathbf{x}_i^{\mathrm{nad}})^2 \leq c\nu^2 \|\boldsymbol{w}\|_2^2 \log(\|\boldsymbol{w}\|_2^2/\delta) \leq c\nu^2 \alpha \log(\alpha/\delta), \tag{27}
$$

where $c$ is a universal constant. Therefore, with probability at least $1 - \delta$,

$$
b_{n,1} \leq \sqrt{d} \cdot \sqrt{c\nu^2 \alpha \log(d\alpha/\delta)}. \tag{28}
$$

**Analysis of $b_{n,2}$:** note $\widehat{\boldsymbol{\theta}}_{\mathrm{nad}}^{\mathrm{Pr}} = \widehat{\boldsymbol{\theta}}_{\mathrm{nad}}^{\mathrm{ols}}$ is the OLS estimate. Therefore, we can use block-wise matrix inverse formula to get its expression. Precisely, we have

$$
\widehat{\boldsymbol{\theta}}_{\mathrm{nad}}^{\mathrm{Pr}} - \boldsymbol{\theta}_{\mathrm{nad}}^* = -\frac{(\mathbf{X}_{\mathrm{nad}}^\top \mathbf{X}_{\mathrm{nad}})^{-1} \mathbf{X}_{\mathrm{nad}}^\top \mathbf{x}_1 \mathbf{x}_1^\top \boldsymbol{\varepsilon}}{\|\mathbf{x}_1 - \mathbf{P}_{\mathbf{X}_{\mathrm{nad}}} \mathbf{x}_1\|_2^2} + (\mathbf{X}_{\mathrm{nad}}^\top \mathbf{X}_{\mathrm{nad}})^{-1} \mathbf{X}_{\mathrm{nad}}^\top \boldsymbol{\varepsilon}
$$

$$
+ \frac{(\mathbf{X}_{\mathrm{nad}}^\top \mathbf{X}_{\mathrm{nad}})^{-1} \mathbf{X}_{\mathrm{nad}}^\top \mathbf{x}_1 \mathbf{x}_1^\top \mathbf{X}_{\mathrm{nad}} (\mathbf{X}_{\mathrm{nad}}^\top \mathbf{X}_{\mathrm{nad}})^{-1} \mathbf{X}_{\mathrm{nad}}^\top \boldsymbol{\varepsilon}}{\|\mathbf{x}_1 - \mathbf{P}_{\mathbf{X}_{\mathrm{nad}}} \mathbf{x}_1\|_2^2}. \tag{29}
$$

Therefore, we can upper bound $b_{n,2}$ by

$$
b_{n,2} \leq \underbrace{\frac{1}{\|\mathbf{x}_1 - \mathbf{P}_{\mathbf{X}_{\mathrm{nad}}} \mathbf{x}_1\|_2^2} \|\|(\mathbf{X}_{\mathrm{nad}}^\top \mathbf{X}_{\mathrm{nad}})^{-1}\|\|_{\mathrm{op}} \cdot \|\mathbf{X}_{\mathrm{nad}}^\top \mathbf{x}_1\|_2 \cdot \|\mathbf{x}_1^\top \boldsymbol{\varepsilon}\|_2}_{:=b_{n,2}^{(1)}}
$$

$$
+ \underbrace{\|(\mathbf{X}_{\mathrm{nad}}^\top \mathbf{X}_{\mathrm{nad}})^{-1} \mathbf{X}_{\mathrm{nad}}^\top \boldsymbol{\varepsilon}\|_2}_{:=b_{n,2}^{(2)}} \tag{30}
$$

$$
+ \underbrace{\frac{1}{\|\mathbf{x}_1 - \mathbf{P}_{\mathbf{X}_{\mathrm{nad}}} \mathbf{x}_1\|_2^2} \|\|(\mathbf{X}_{\mathrm{nad}}^\top \mathbf{X}_{\mathrm{nad}})^{-1}\|\|_{\mathrm{op}} \cdot \|\mathbf{X}_{\mathrm{nad}}^\top \mathbf{x}_1\|_2^2 \cdot \|(\mathbf{X}_{\mathrm{nad}}^\top \mathbf{X}_{\mathrm{nad}})^{-1} \mathbf{X}_{\mathrm{nad}}^\top \boldsymbol{\varepsilon}\|_2}_{:=b_{n,2}^{(3)}}.
$$

To analyze terms $b_{n,2}^{(1)}, b_{n,2}^{(2)}$, and $b_{n,2}^{(3)}$, we make use of the results from Lemma A.1, Lemma A.2, and Lemma A.3. Specifically, we have with probability $1 - \delta$, the following statements hold

$$
\begin{aligned}
b_{n,2}^{(1)} &= \frac{1}{\|\mathbf{x}_1 - \mathbf{P}_{\mathbf{X}_{\mathrm{nad}}}\mathbf{x}_1\|_2^2} \cdot \|\|(\mathbf{X}_{\mathrm{nad}}^\top \mathbf{X}_{\mathrm{nad}})^{-1}\|\|_{\mathrm{op}} \cdot \|\mathbf{X}_{\mathrm{nad}}^\top \mathbf{x}_1\|_2 \cdot \|\mathbf{x}_1^\top \boldsymbol{\varepsilon}\|_2 \\
&\overset{(i)}{\leq} c' \left( \frac{1}{(1 - Cd\log(n/\delta)/n) \cdot \|\mathbf{x}_1\|_2^2} \right) \cdot \frac{1}{n} \cdot \|\mathbf{x}_1\|_2 \sqrt{d}\sqrt{\log(d\|\mathbf{x}_1\|_2^2/\delta)} \\
&\qquad \cdot \|\mathbf{x}_1\|_2 \sqrt{\log(\|\mathbf{x}_1\|_2^2/\delta)} \\
&\leq \frac{c'\sqrt{d}\log(n^2 d/\delta)}{(1 - Cd\log(n/\delta)/n)n} \\
b_{n,2}^{(2)} &= \|(\mathbf{X}_{\mathrm{nad}}^\top \mathbf{X}_{\mathrm{nad}})^{-1}\mathbf{X}_{\mathrm{nad}}^\top \boldsymbol{\varepsilon}\|_2 \leq \|\|(\mathbf{X}_{\mathrm{nad}}^\top \mathbf{X}_{\mathrm{nad}})^{-1}\|\|_{\mathrm{op}} \cdot \|\mathbf{X}_{\mathrm{nad}}^\top \boldsymbol{\varepsilon}\|_2 \quad (31) \\
&\overset{(ii)}{\leq} c'' \sqrt{\frac{d\log(d/\delta)}{n}} \\
b_{n,2}^{(3)} &= \frac{1}{\|\mathbf{x}_1 - \mathbf{P}_{\mathbf{X}_{\mathrm{nad}}}\mathbf{x}_1\|_2^2} \|\|(\mathbf{X}_{\mathrm{nad}}^\top \mathbf{X}_{\mathrm{nad}})^{-1}\|\|_{\mathrm{op}} \cdot \|\mathbf{X}_{\mathrm{nad}}^\top \mathbf{x}_1\|_2^2 \cdot \|(\mathbf{X}_{\mathrm{nad}}^\top \mathbf{X}_{\mathrm{nad}})^{-1}\mathbf{X}_{\mathrm{nad}}^\top \boldsymbol{\varepsilon}\|_2 \\
&\overset{(iii)}{\leq} \frac{1}{(1 - Cd\log(n/\delta)/n) \cdot \|\mathbf{x}_1\|_2^2} \cdot \frac{c'''}{n} \cdot d\|\mathbf{x}_1\|_2^2 \log(d\|\mathbf{x}_1\|_2^2/\delta) \cdot \sqrt{\frac{d\log(d/\delta)}{n}} \\
&\leq c''' \frac{1}{(1 - Cd\log(n/\delta)/n)} \frac{d^{3/2}\{\log(dn^2/\delta)\}^{3/2}}{n^{3/2}},
\end{aligned}
$$

where $c'$, $c''$ and $c'''$ are universal constants that are independent of $n$ and $d$. In inequality $(i)$, we use Lemma A.3 to obtain a lower bound for $\|\mathbf{x}_1 - \mathbf{P}_{\mathbf{X}_{\mathrm{nad}}}\mathbf{x}_1\|_2^2$ and apply Lemma A.1 and Lemma A.2 to control the other three terms separately. Inequality $(ii)$ makes use of the fact that $\boldsymbol{x}_{ij}\varepsilon_i$ is sub-exponential with parameter $(c\nu v, c\nu v)$ conditioned on $\mathcal{F}_{i-1}$ for $j = 2, \ldots, d$. Therefore, by Azuma-Bernstein inequality and the sample size assumption, we obtain for $2 \leq j \leq d$,

$$
|\mathbf{x}_j^\top \boldsymbol{\varepsilon}| \leq \nu v \sqrt{\log(d/\delta)}\left( \sqrt{n} \vee \sqrt{\log(d/\delta)} \right) = \nu v \sqrt{n\log(d/\delta)}
$$

with probability over $1 - \delta/d$ for any $2 \leq j \leq d$. Applying a union bound to $j = 2, \ldots, d$, we have

$$
\|\mathbf{X}_{\mathrm{nad}}^\top \boldsymbol{\varepsilon}\|_2 \leq C'nd\log(d/\delta), \quad (32)
$$

for some constant $C'$. Inequality $(iii)$ makes use of the bound for $b_{n,2}^{(2)}$ and Lemmas A.1 and A.2. Therefore, when $d^2 \log^2(n)/n \to 0$, we conclude

$$
b_{n,1} \cdot b_{n,2} = o_p(1). \quad (33)
$$

With $b_n \overset{p}{\to} 0$ at hand, a direct application of Slutsky's theorem yields

$$
\frac{1}{\widehat{\sigma}\sqrt{\sum_{1 \leq i \leq n} w_i^2}}\left( \sum_{i=1}^n w_i \boldsymbol{x}_i^{\mathrm{nad}\top} \right) \cdot (\widehat{\boldsymbol{\theta}}_{\mathrm{nad}}^{\mathrm{Pr}} - \boldsymbol{\theta}_{\mathrm{nad}}^*) \overset{p}{\to} 0. \quad (34)
$$

Putting things together, we conclude that

$$
\frac{1}{\widehat{\sigma}\sqrt{\sum_{1 \leq i \leq n} w_i^2}}\left( \sum_{i=1}^n w_i x_i^{\mathrm{ad}} \right) \cdot (\widehat{\theta}_{\mathrm{TALE}} - \theta_1^*) \overset{d}{\to} \mathcal{N}(0, 1). \quad (35)
$$

## A.7 Proof of Lemma A.1

**Proof of Part (a)** The proof follows immediately from choosing $V = \mathbf{I}_d$ in Theorem 1 of Abbasi et al. [1].

**Proof of Part (b)** Define $\bar{\mathbf{S}}_1 := \widetilde{\mathbf{A}}^\top \widetilde{\mathbf{A}}$ and $\mathbf{A}_{\mathrm{aug}} := [\mathbf{A} \quad \mathbf{1}_n]$. Then

$$
\begin{aligned}
\mathbf{b}^\top \mathbf{P}_{\widetilde{\mathbf{A}}} \mathbf{b} &= \mathbf{b}^\top \widetilde{\mathbf{A}} \bar{\mathbf{S}}_1^{-1} \bar{\mathbf{S}}_1 \bar{\mathbf{S}}_1^{-1} \widetilde{\mathbf{A}}^\top \mathbf{b} \\
&= \mathbf{b}^\top \mathbf{A}_{\mathrm{aug}} (\mathbf{A}_{\mathrm{aug}}^\top \mathbf{A}_{\mathrm{aug}})^{-1} \begin{pmatrix} \mathbf{I}_k \\ 0 \end{pmatrix} (\mathbf{I}_k \quad 0) \bar{\mathbf{S}}_1 \begin{pmatrix} \mathbf{I}_k \\ 0 \end{pmatrix} (\mathbf{I}_k \quad 0) (\mathbf{A}_{\mathrm{aug}}^\top \mathbf{A}_{\mathrm{aug}})^{-1} \mathbf{A}_{\mathrm{aug}}^\top \mathbf{b} \\
&\leq \mathbf{b}^\top \mathbf{A}_{\mathrm{aug}} (\mathbf{A}_{\mathrm{aug}}^\top \mathbf{A}_{\mathrm{aug}})^{-1} (\mathbf{A}_{\mathrm{aug}}^\top \mathbf{A}_{\mathrm{aug}}) (\mathbf{A}_{\mathrm{aug}}^\top \mathbf{A}_{\mathrm{aug}})^{-1} \mathbf{A}_{\mathrm{aug}}^\top \mathbf{b} \\
&= \mathbf{b}^\top \mathbf{A}_{\mathrm{aug}} (\mathbf{A}_{\mathrm{aug}}^\top \mathbf{A}_{\mathrm{aug}})^{-1} \mathbf{A}_{\mathrm{aug}}^\top \mathbf{b} \\
&\leq c\sigma^2 \log(\det(\mathbf{A}_{\mathrm{aug}}^\top \mathbf{A}_{\mathrm{aug}})/\delta) \\
&= c\sigma^2 \log(n \det(\widetilde{\mathbf{A}}^\top \widetilde{\mathbf{A}})/\delta)
\end{aligned}
$$

for all $k+1 \leq j \leq d$ with probability over $1 - \delta$. Here the first equality comes from the definition of $\mathbf{P}$; the second equality uses the fact that the coefficients of $\mathbf{A}$ in the ordinary least squares (OLS) estimator for the linear model $\mathbf{b} \sim \mathbf{A} + \mathbf{1}$ equals the OLS estimator for the centered linear model $\mathbf{b} \sim \widetilde{\mathbf{A}}$, i.e.,

$$
\begin{pmatrix} \mathbf{I}_k \\ 0 \end{pmatrix} (\mathbf{I}_k \quad 0) (\mathbf{A}_{\mathrm{aug}}^\top \mathbf{A}_{\mathrm{aug}})^{-1} \mathbf{A}_{\mathrm{aug}}^\top \mathbf{b} = \bar{\mathbf{S}}_1^{-1} (\mathbf{A} - \mathbf{P}_{\mathbf{1}_n} \mathbf{A})^\top \mathbf{b};
$$

the third line is due to the fact that $\begin{bmatrix} \bar{\mathbf{S}}_1 & \mathbf{0}_k \\ \mathbf{0}_k^\top & 0 \end{bmatrix} \preceq \mathbf{A}_{\mathrm{aug}}^\top \mathbf{A}_{\mathrm{aug}}$; the fifth line follows from Lemma A.1 and the last line exploits the Schur complement of $\widetilde{\mathbf{A}}^\top \widetilde{\mathbf{A}}$.

### A.8 Proof of Lemma A.2

Let $\boldsymbol{\mu}^* = (\boldsymbol{\mu}_{\mathrm{ad}}^{*\top}, \boldsymbol{\mu}_{\mathrm{nad}}^{*\top})^\top$ denote the mean vector of $\mathbb{E}[\boldsymbol{x}_i]$ and define

$$
\overline{\mathbf{X}}_{\mathrm{nad}} := \mathbf{X}_{\mathrm{nad}} - \mathbf{1}_n \boldsymbol{\mu}_{\mathrm{nad}}^{*\top}.
$$

**Proof of** (14a) **and** (14b). Applying the concentration result for a sample covariance matrix of sub-Gaussian ensemble (see e.g., Theorem 6.5 in Wainwright [37]), we obtain

$$
\|\overline{\mathbf{X}}_{\mathrm{nad}}^\top \overline{\mathbf{X}}_{\mathrm{nad}} - n\boldsymbol{\Sigma}\|_{\mathrm{op}} \leq c\nu^2 n \Big( \sqrt{\frac{d-k+\log(1/\delta)}{n}} + \frac{d-k+\log(1/\delta)}{n} \Big)
$$

with probability over $1 - \delta$. Using Weyl's theorem (see e.g., Theorem 4.3.1 in Horn et al. [18]), the sample size assumption and the last display, we find that

$$
\sigma_{\min}(\overline{\mathbf{X}}_{\mathrm{nad}}^\top \overline{\mathbf{X}}_{\mathrm{nad}}) \geq n\sigma_{\min}(\boldsymbol{\Sigma}) - \|\overline{\mathbf{X}}_{\mathrm{nad}}^\top \overline{\mathbf{X}}_{\mathrm{nad}} - n\boldsymbol{\Sigma}\|_{\mathrm{op}} \geq \frac{n\sigma_{\min}}{2}.
$$

Similarly, we have $\|\overline{\mathbf{X}}_{\mathrm{nad}}\|_{\mathrm{op}} = (\sigma_{\max}(\overline{\mathbf{X}}_{\mathrm{nad}}^\top \overline{\mathbf{X}}_{\mathrm{nad}}))^{1/2} \leq \sqrt{n(\sigma_{\max} + \sigma_{\min}/2)} \leq \sqrt{2n\sigma_{\max}}$. This concludes (14a), (14b).

**Proof of** (14c). Recall that we use $\hat{\boldsymbol{\mu}} = (\hat{\boldsymbol{\mu}}_{\mathrm{ad}}^{*\top}, \hat{\boldsymbol{\mu}}_{\mathrm{nad}}^{*\top})^\top$ to denote the empirical average of $\{\boldsymbol{x}_i\}_{i=1}^n$. From Assumption (A2) and properties of sub-Gaussian vectors, we have with probability over $1 - \delta$

$$
\|\hat{\boldsymbol{\mu}}_{\mathrm{nad}} - \boldsymbol{\mu}_{\mathrm{nad}}^*\|_2 \leq c\nu \sqrt{\log((d-k)/\delta)} \sqrt{\frac{d-k}{n}}. \tag{36}
$$

It follows immediately that

$$
\|\overline{\mathbf{X}}_{\mathrm{nad}} - \widetilde{\mathbf{X}}_{\mathrm{nad}}\|_{\mathrm{op}} = \|\mathbf{1}_n(\hat{\boldsymbol{\mu}}_{\mathrm{nad}} - \boldsymbol{\mu}_{\mathrm{nad}}^*)^\top\|_{\mathrm{op}} = \|\hat{\boldsymbol{\mu}}_{\mathrm{nad}} - \boldsymbol{\mu}_{\mathrm{nad}}^*\|_2 \|\mathbf{1}_n\|_2 \leq C\sqrt{\log(n/\delta)} \sqrt{d-k}.
$$

The second inequality in (14c) follows from equation (14a) and the sample size assumption 6.

**Proof of equation** (14d) **and** (14e). By Woodbury's matrix identity, we have

$$
\begin{aligned}
& \|(\overline{\mathbf{X}}_{\mathrm{nad}}^\top \overline{\mathbf{X}}_{\mathrm{nad}})^{-1} - (\widetilde{\mathbf{X}}_{\mathrm{nad}}^\top \widetilde{\mathbf{X}}_{\mathrm{nad}})^{-1}\|_{\mathrm{op}} \\
& \leq \|(\overline{\mathbf{X}}_{\mathrm{nad}}^\top \overline{\mathbf{X}}_{\mathrm{nad}})^{-1}\|_{\mathrm{op}} \|\overline{\mathbf{X}}_{\mathrm{nad}}^\top \overline{\mathbf{X}}_{\mathrm{nad}} - \widetilde{\mathbf{X}}_{\mathrm{nad}}^\top \widetilde{\mathbf{X}}_{\mathrm{nad}}\|_{\mathrm{op}} \|(\widetilde{\mathbf{X}}_{\mathrm{nad}}^\top \widetilde{\mathbf{X}}_{\mathrm{nad}})^{-1}\|_{\mathrm{op}}.
\end{aligned}
$$

By equation (14a), (14b), (14c) and a standard triangular inequality, we find

$$\|\overline{\mathbf{X}}_{\mathrm{nad}}^\top \overline{\mathbf{X}}_{\mathrm{nad}} - \widetilde{\mathbf{X}}_{\mathrm{nad}}^\top \widetilde{\mathbf{X}}_{\mathrm{nad}}\|_{\mathrm{op}} \le C\sqrt{\log(n/\delta)}\sqrt{(d-k)n} \le \frac{n\sigma_{\min}}{4} \le \frac{1}{2}\sigma_{\min}(\overline{\mathbf{X}}_{\mathrm{nad}}^\top \overline{\mathbf{X}}_{\mathrm{nad}}),$$

where the second inequality follows from the sample size assumption (6). Therefore, we have

$$\|(\widetilde{\mathbf{X}}_{\mathrm{nad}}^\top \widetilde{\mathbf{X}}_{\mathrm{nad}})^{-1}\|_{\mathrm{op}} \le \frac{2}{\sigma_{\min}(\overline{\mathbf{X}}_{\mathrm{nad}}^\top \overline{\mathbf{X}}_{\mathrm{nad}})}$$

and hence

$$\|(\overline{\mathbf{X}}_{\mathrm{nad}}^\top \overline{\mathbf{X}}_{\mathrm{nad}})^{-1} - (\widetilde{\mathbf{X}}_{\mathrm{nad}}^\top \widetilde{\mathbf{X}}_{\mathrm{nad}})^{-1}\|_{\mathrm{op}} \le \frac{C}{\sigma_{\min}(\overline{\mathbf{X}}_{\mathrm{nad}}^\top \overline{\mathbf{X}}_{\mathrm{nad}})^2}\|\overline{\mathbf{X}}_{\mathrm{nad}}^\top \overline{\mathbf{X}}_{\mathrm{nad}} - \widetilde{\mathbf{X}}_{\mathrm{nad}}^\top \widetilde{\mathbf{X}}_{\mathrm{nad}}\|_{\mathrm{op}}$$

$$\le \frac{C\sqrt{\log(n/\delta)}\sqrt{d-k}}{n^{3/2}}. \tag{37}$$

This gives equation (14d). Moreover, note that

$$\|\mathbf{P}_{\overline{\mathbf{X}}_{\mathrm{nad}}} - \mathbf{P}_{\widetilde{\mathbf{X}}_{\mathrm{nad}}}\|_{\mathrm{op}}$$
$$\le \|\widetilde{\mathbf{X}}_{\mathrm{nad}}[(\overline{\mathbf{X}}_{\mathrm{nad}}^\top \overline{\mathbf{X}}_{\mathrm{nad}})^{-1} - (\widetilde{\mathbf{X}}_{\mathrm{nad}}^\top \widetilde{\mathbf{X}}_{\mathrm{nad}})^{-1}]\widetilde{\mathbf{X}}_{\mathrm{nad}}^\top\|_{\mathrm{op}}$$
$$\quad + \|(\overline{\mathbf{X}}_{\mathrm{nad}} - \widetilde{\mathbf{X}}_{\mathrm{nad}})(\overline{\mathbf{X}}_{\mathrm{nad}}^\top \overline{\mathbf{X}}_{\mathrm{nad}})^{-1}(\overline{\mathbf{X}}_{\mathrm{nad}} - \widetilde{\mathbf{X}}_{\mathrm{nad}})^\top\|_{\mathrm{op}}$$
$$\quad + 2\|(\overline{\mathbf{X}}_{\mathrm{nad}} - \widetilde{\mathbf{X}}_{\mathrm{nad}})^\top(\overline{\mathbf{X}}_{\mathrm{nad}}^\top \overline{\mathbf{X}}_{\mathrm{nad}})^{-1}\overline{\mathbf{X}}_{\mathrm{nad}}^\top\|_{\mathrm{op}}$$
$$\le \|\widetilde{\mathbf{X}}_{\mathrm{nad}}\|_{\mathrm{op}}^2 \|(\overline{\mathbf{X}}_{\mathrm{nad}}^\top \overline{\mathbf{X}}_{\mathrm{nad}})^{-1} - (\widetilde{\mathbf{X}}_{\mathrm{nad}}^\top \widetilde{\mathbf{X}}_{\mathrm{nad}})^{-1}\|_{\mathrm{op}}$$
$$\quad + \|\overline{\mathbf{X}}_{\mathrm{nad}} - \widetilde{\mathbf{X}}_{\mathrm{nad}}\|_{\mathrm{op}}(\|\overline{\mathbf{X}}_{\mathrm{nad}} - \widetilde{\mathbf{X}}_{\mathrm{nad}}\|_{\mathrm{op}} + 2\|\overline{\mathbf{X}}_{\mathrm{nad}}\|_{\mathrm{op}})\|(\overline{\mathbf{X}}_{\mathrm{nad}}^\top \overline{\mathbf{X}}_{\mathrm{nad}})^{-1}\|_{\mathrm{op}}.$$

It follows immediately from equation (14a), (14b), (14c), (14d) and (37) that with probability over $1-\delta$

$$\|\mathbf{P}_{\overline{\mathbf{X}}_{\mathrm{nad}}} - \mathbf{P}_{\widetilde{\mathbf{X}}_{\mathrm{nad}}}\|_{\mathrm{op}} \le \frac{C\sqrt{\log(n/\delta)}\sqrt{d-k}}{\sqrt{n}}.$$

This yields equation (14e).

**Proof of equation** (14f). Define $\boldsymbol{\Delta} := \hat{\boldsymbol{\mu}} - \boldsymbol{\mu}^*$. We have

$$\|(\mathbf{P}_{\overline{\mathbf{X}}_{\mathrm{nad}}} - \mathbf{P}_{\widetilde{\mathbf{X}}_{\mathrm{nad}}})\boldsymbol{\varepsilon}\|_2$$
$$\le \|\widetilde{\mathbf{X}}_{\mathrm{nad}}(\widetilde{\mathbf{X}}_{\mathrm{nad}}^\top \widetilde{\mathbf{X}}_{\mathrm{nad}})^{-1}\boldsymbol{\Delta}\mathbf{1}_n^\top \boldsymbol{\varepsilon}\|_2 + \|(\widetilde{\mathbf{X}}_{\mathrm{nad}}(\widetilde{\mathbf{X}}_{\mathrm{nad}}^\top \widetilde{\mathbf{X}}_{\mathrm{nad}})^{-1} - \overline{\mathbf{X}}_{\mathrm{nad}}(\overline{\mathbf{X}}_{\mathrm{nad}}^\top \overline{\mathbf{X}}_{\mathrm{nad}})^{-1})\overline{\mathbf{X}}_{\mathrm{nad}}^\top \boldsymbol{\varepsilon}\|_2$$
$$\le \|\widetilde{\mathbf{X}}_{\mathrm{nad}}(\widetilde{\mathbf{X}}_{\mathrm{nad}}^\top \widetilde{\mathbf{X}}_{\mathrm{nad}})^{-1}\|_{\mathrm{op}}\|\boldsymbol{\Delta}\|_2|\mathbf{1}_n^\top \boldsymbol{\varepsilon}| + \|\widetilde{\mathbf{X}}_{\mathrm{nad}}(\widetilde{\mathbf{X}}_{\mathrm{nad}}^\top \widetilde{\mathbf{X}}_{\mathrm{nad}})^{-1}$$
$$\quad - \overline{\mathbf{X}}_{\mathrm{nad}}(\overline{\mathbf{X}}_{\mathrm{nad}}^\top \overline{\mathbf{X}}_{\mathrm{nad}})^{-1}\|_{\mathrm{op}}\|\overline{\mathbf{X}}_{\mathrm{nad}}^\top \overline{\mathbf{X}}_{\mathrm{nad}}\|_{\mathrm{op}}^{1/2}\|\widetilde{\mathbf{P}}_{\overline{\mathbf{X}}_{\mathrm{nad}}}\boldsymbol{\varepsilon}\|_2.$$

Since $|\mathbf{1}_n^\top \boldsymbol{\varepsilon}| \le cv\sqrt{n\log(1/\delta)}$ with probability over $1-\delta$ by Assumption (A4) and concentration of sub-Gaussian variables, and

$$\|\widetilde{\mathbf{X}}_{\mathrm{nad}}(\widetilde{\mathbf{X}}_{\mathrm{nad}}^\top \widetilde{\mathbf{X}}_{\mathrm{nad}})^{-1} - \overline{\mathbf{X}}_{\mathrm{nad}}(\overline{\mathbf{X}}_{\mathrm{nad}}^\top \overline{\mathbf{X}}_{\mathrm{nad}})^{-1}\|_{\mathrm{op}}$$
$$\le \|\widetilde{\mathbf{X}}_{\mathrm{nad}} - \overline{\mathbf{X}}_{\mathrm{nad}}\|_{\mathrm{op}}\|(\widetilde{\mathbf{X}}_{\mathrm{nad}}^\top \widetilde{\mathbf{X}}_{\mathrm{nad}})^{-1}\|_{\mathrm{op}}$$
$$\quad + \|\overline{\mathbf{X}}_{\mathrm{nad}}\|_{\mathrm{op}}\|(\widetilde{\mathbf{X}}_{\mathrm{nad}}^\top \widetilde{\mathbf{X}}_{\mathrm{nad}})^{-1} - (\overline{\mathbf{X}}_{\mathrm{nad}}^\top \overline{\mathbf{X}}_{\mathrm{nad}})^{-1}\|_{\mathrm{op}}$$

by the triangular inequality, it follows immediately from combining equation (14a), (14b), (14c), (14d), (36), (37) that

$$\|(\mathbf{P}_{\overline{\mathbf{X}}_{\mathrm{nad}}} - \mathbf{P}_{\widetilde{\mathbf{X}}_{\mathrm{nad}}})\boldsymbol{\varepsilon}\|_2 \le C\Big[\sqrt{\frac{(d-k)\log(n/\delta)\log(1/\delta)}{n}} + \sqrt{\frac{(d-k)^2\log^2(n/\delta)}{n}}\Big] \le C$$

with probability over $1-\delta$.

## A.9 Proof of Lemma A.3

Denote $(\mathbf{X}_{\mathrm{ad}}^\top \mathbf{X}_{\mathrm{ad}})^{-1/2}\mathbf{X}_{\mathrm{ad}}^\top \mathbf{P}_{\mathbf{X}_{\mathrm{nad}}}\mathbf{X}_{\mathrm{ad}}(\mathbf{X}_{\mathrm{ad}}^\top \mathbf{X}_{\mathrm{ad}})^{-1/2}$ by $\mathbf{B}$. By our construction and noting that $\|\mathbf{B}\|_{\mathrm{op}} \leq \mathrm{tr}(\mathbf{B})$, it suffices to show

$$\mathrm{tr}(\mathbf{B}) \leq \frac{C(d-k)k\log(n/\delta)}{n}$$

with probability over $1 - \delta$ for some $C > 0$. Plugging in the definition of $\mathbf{P}_{\mathbf{X}_{\mathrm{nad}}}$, we obtain

$$
\begin{aligned}
\mathrm{tr}(\mathbf{B}) &= \mathrm{tr}((\mathbf{X}_{\mathrm{ad}}^\top \mathbf{X}_{\mathrm{ad}})^{-1/2}\mathbf{X}_{\mathrm{ad}}^\top \mathbf{X}_{\mathrm{nad}}(\mathbf{X}_{\mathrm{nad}}^\top \mathbf{X}_{\mathrm{nad}})^{-1}\mathbf{X}_{\mathrm{nad}}^\top \mathbf{X}_{\mathrm{ad}}(\mathbf{X}_{\mathrm{ad}}^\top \mathbf{X}_{\mathrm{ad}})^{-1/2}) \\
&= \mathrm{tr}(\mathbf{X}_{\mathrm{nad}}^\top \mathbf{X}_{\mathrm{ad}}(\mathbf{X}_{\mathrm{ad}}^\top \mathbf{X}_{\mathrm{ad}})^{-1}\mathbf{X}_{\mathrm{ad}}^\top \mathbf{X}_{\mathrm{nad}}(\mathbf{X}_{\mathrm{nad}}^\top \mathbf{X}_{\mathrm{nad}})^{-1}) \\
&\leq \mathrm{tr}(\mathbf{X}_{\mathrm{nad}}^\top \mathbf{X}_{\mathrm{ad}}(\mathbf{X}_{\mathrm{ad}}^\top \mathbf{X}_{\mathrm{ad}})^{-1}\mathbf{X}_{\mathrm{ad}}^\top \mathbf{X}_{\mathrm{nad}}) \cdot \|(\mathbf{X}_{\mathrm{nad}}^\top \mathbf{X}_{\mathrm{nad}})^{-1}\|_{\mathrm{op}} \\
&\leq \frac{c}{n\sigma_{\min}} \mathrm{tr}(\mathbf{X}_{\mathrm{nad}}^\top \mathbf{X}_{\mathrm{ad}}(\mathbf{X}_{\mathrm{ad}}^\top \mathbf{X}_{\mathrm{ad}})^{-1}\mathbf{X}_{\mathrm{ad}}^\top \mathbf{X}_{\mathrm{nad}})
\end{aligned}
\tag{38}
$$

with probability over $1 - \delta$, where the third line follows from von Neumann's trace inequality (see e.g., Theorem A.15 in Bai et al. [3]), and the last line uses equation (14a). Write $\mathbf{X}_{\mathrm{ad}} = [\mathbf{x}_1, \mathbf{x}_2, \ldots, \mathbf{x}_k] \in \mathbb{R}^{n \times k}$ and $\mathbf{X}_{\mathrm{nad}} = [\mathbf{x}_{k+1}, \mathbf{x}_{k+2}, \ldots, \mathbf{x}_d] \in \mathbb{R}^{n \times (d-k)}$. Following the calculation, we further have

$$\mathrm{tr}(\mathbf{X}_{\mathrm{nad}}^\top \mathbf{X}_{\mathrm{ad}}(\mathbf{X}_{\mathrm{ad}}^\top \mathbf{X}_{\mathrm{ad}})^{-1}\mathbf{X}_{\mathrm{ad}}^\top \mathbf{X}_{\mathrm{nad}}) = \sum_{j=k+1}^{d} \mathbf{x}_j^\top \mathbf{P}_{\mathbf{X}_{\mathrm{ad}}}\mathbf{x}_j. \tag{39}$$

It follows from Lemma A.1 that

$$\mathbf{x}_j^\top \mathbf{P}_{\mathbf{X}_{\mathrm{ad}}}\mathbf{x}_j \leq Ck\log(n/\delta) \tag{40}$$

for all $k + 1 \leq j \leq d$ with probability over $1 - \delta$. The desired result follows immediately from combining equation (40) with (38) and (39).

## A.10 Proof of Lemma A.4

Following the same arguments as in the proof of Lemma A.3 with $\mathbf{X}_{\mathrm{ad}}$ replaced by $\widetilde{\mathbf{X}}_{\mathrm{ad}} := \mathbf{X}_{\mathrm{ad}} - \mathbf{P}_{\mathbf{1}_n}\mathbf{X}_{\mathrm{ad}}$, it suffices to show

$$\mathbf{x}_j^\top \mathbf{P}_{\widetilde{\mathbf{X}}_{\mathrm{ad}}}\mathbf{x}_j \leq Ck\log(n/\delta). \tag{41}$$

for all $k + 1 \leq j \leq d$ with probability over $1 - \delta$. This follows immediately from Lemma A.1 (b).

# B Simulation set up

We provide implementation details of our simulations in this section. The code is available at `https://github.com/licong-lin/low-dim-debias`.

## B.1 Single coordinate estimation

In this section, we detail the simulation set up that was used to generate the Figure 1. The goal of this simulation is to display the effect of degree of adaptivity $k$ on the estimation of a single coordinate, and provide empirical validation to the theory developed in the paper (c.f. Corollary 3.3).

We want to design an adaptive data collection mechanism, which can capture the estimation lower bound. Therefore, we adopt a similar data collection procedure as provided in Khamaru et al. [20]. We also refer readers to Lattimore [25] for related information.

**Simulation set-up:**

- Sample size $n = 1000$, $d = 300$.
- The degree of adaptivity $k$ varies from 2 to 200 with step size equal to 3.
- Replication number 20 for each level of adaptivity $(k, d)$.
- $\theta_1^{\mathrm{ad}} = 1$ and other coefficients are generated independently from $\mathcal{N}(0, 1)$.
- $\boldsymbol{x}_i^{\mathrm{nad}}$ is generated independently from uniform distribution on the sphere $\mathcal{S}^{d-k-1}$. If $\boldsymbol{x}_i^{\mathrm{nad}}$ has mean not equal to zero, then we consider $\boldsymbol{x}_i^{\mathrm{nad}}$ plus $\mathbb{E}\boldsymbol{x}_i^{\mathrm{nad}}$, where $\mathbb{E}\boldsymbol{x}_i^{\mathrm{nad}}$ is generated from $\mathcal{N}(\mathbf{1}, \mathbf{I}_{d-k})$.

**Data collection method:** Here we modified the data collection algorithm from Section 5.2.2 in Khamaru et al. [20]. The only difference between our data collection algorithm and the one in Khamaru et al. [20] is that we replace $m_{u,v} := \sum_{w=1}^{v} b_w (y_{u,w} - a_{u,w})$ by

$$m_{u,v} := \sum_{w=1}^{v} b_w \left( y_{u,w} - a_{u,w} - \boldsymbol{\theta}^{\text{nad}\top} \boldsymbol{x}_{u+(w-1)(d-1)}^{\text{nad}} \right). \tag{42}$$

Figure 1 shows that empirical relation between the MSE of the centered OLS estimate of the first coordinate and the degree of dependence $k$.

## B.2   Single coordinate inference

In this section, we detail the simulation set up that is used to generate the Figure 2 and 3.

We generate a dataset $\{(\boldsymbol{x}_i, y_i)\}_{i=1}^{n}$ that satisfies the assumptions in Theorem 3.4. On this simulated dataset, we compare our method with the ordinary least squares (OLS) estimator, W-decorrelation proposed by Deshpande [13], and the non-asymptotic confidence intervals derived from Theorem 8 in Lattimore et al. [26].

We begin by describing our data generating mechanism. We assume the data $\{(\boldsymbol{x}_i, y_i)\}_{i=1}^{n} \in \mathbb{R}^d \times \mathbb{R}$ are generated from a linear model $y_i = \boldsymbol{x}_i^\top \boldsymbol{\theta}^* + \varepsilon_i$, where $\varepsilon_i \overset{iid}{\sim} \mathcal{N}(0, \sigma^2)$. We generate the covariates $\{\boldsymbol{x}_i\}_{i=1}^{n}$ in the following way

1. We assume the non-adaptive component $\boldsymbol{x}_{i,2:d}$ are i.i.d $\mathcal{N}(0, \mathbf{I}_{d-1})$ across $i \in [n]$.

2. For the adaptive coordinate, we choose $x_{1,1} = 1$ and assume $x_{i,1} \in \{0, 1\}$ for all $i \in [n]$.

3. At each stage $i \geq 2$, denote by $\widehat{\theta}_1^{(i)}$ the OLS estimator for the first coordinate $\theta_1^*$ obtained using the first $i - 1$ samples $(\mathbf{X}_{1:i-1}, \boldsymbol{y}_{1:i-1})$. With probability $p$ we choose $x_{i,1} = 1$ if $\widehat{\theta}_1^{(i)} > 0$ and $x_{i,1} = 0$ if otherwise; with probability $1 - p$ we simply choose $x_{i,1} = 1$ to encourage exploration.

Recalling Example 2.1 on treatment assignment, in the simulated data, we use the OLS estimator to obtain an prior estimate of the treatment effect $\theta_1^*$ and assign the treatment to the $i$-th patient if the prior estimation suggests that the treatment has a positive effect $(i.e., \widehat{\theta}_1^{(i)} > 0)$. Moreover, to encourage exploration, we assign the treatment (i.e., $x_{i,1} = 1$) with some small probability $1 - p$, regardless of the prior estimation.

Throughout the simulation we choose $\boldsymbol{\theta}_{2:d}^* = \mathbf{1}_{d-1}/\sqrt{d-1}$ and $\theta_1^* = 0$, which corresponds to the case where no treatment effect is presented. We choose the noise level $\sigma = 0.3$ and the probability $p = 0.8$. In the simulations we assume the noise level is known for simplicity. We run our simulations on both a low-dimensional model ($n = 1000, d = 10$) and a high-dimensional model ($n = 500, d = 50$).

**Comparison with W-decorrelation by Deshpande et al. [13]**   In our implementation of W-decorrelation, we follow Algorithm 1 in [13], with the parameter $\lambda \cdot \log(n)$ be the $1/n$-quantile of $\sigma_{\min}(\mathbf{X}^\top \mathbf{X})$. To estimate the quantile, we use the sample estimate from 1000 i.i.d. data matrices $\mathbf{X}$'s to estimate the quantile.

