# OpenReview forum: "Statistical Limits of Adaptive Linear Models: Low-Dimensional Estimation and Inference"
_NeurIPS.cc/2023/Conference — NeurIPS 2023 poster_

### Official Review · Reviewer_KiDv · 2023-07-01

**Soundness:** 3 good
**Presentation:** 3 good
**Contribution:** 2 fair
**Rating:** 5
**Confidence:** 3

**Summary:**

The paper studies the statistical limits of some adaptive linear models.

The paper defines the notion of $(k,d)$-adaptivity (Definition 2.1), and then it proves, under some conditions and for a $(k,d)$-adaptive model and failure probability $\delta$, that
- (Theorem 3.1) the estimation error of the adaptive components is bounded by $k\log(n/\delta)$ if the non-adaptive component has zero mean,
- (Theorem 3.2) and similar results hold if the mean of the non-adaptive component is not zero;
- (Theorem 3.4) moreover, there exists an estimator (TALE) that enjoys asymptotic normality.

The experiments present an interesting phenomenon: The TALE estimator coincides well with the normal distribution.



**Strengths:**

The paper is written clearly and the presentation is smooth and relatively easy to follow.






**Weaknesses:**

I have a major concern about whether the paper is technically solid and I wish to read the response from the authors:
- Regarding Theorems 3.1 and 3.2, could the authors justify how the proof techniques differ from prior works? It appears to me that the two results are basic extensions of prior proofs (so, correct me if I am wrong).
- Theorem 3.4 holds only for a single adaptive coordinate. Could the authors elaborate on the difficulty of extending the results for general $(k,d)$-adaptivity?


Minor:
- the same symbol $\sigma$ is used to represent sigma field, variance of a random variable, and singular values. In my humble opinion, this might be confusing.


**NOTE**: I am not an expert on statistics in general and on this particular line of research, so it is not for me to say whether the paper is significant or novel.

**Questions:**

- From the paper it is a bit hard for me to understand what "adaptivity" precisely means. The definition of $(k,d)$-adaptivity only specifies that the adaptive components $x_i^{ad}$ depend on the sigma field $F_{i-1}$. Would it be much simpler and clearer to say "dependency" instead of "adaptivity"?


**Limitations:**

The authors discussed future works and limitations at the end of the paper.

---

> ### Author Rebuttal · Authors · 2023-08-02
>
> We would like to thank the reviewer for taking the time to review our paper and provide helpful feedback! We truly appreciate your comments and suggestions, and believe they can make our work better.
>
> In the following, we hope to address each of the points made in the review, in the order as they came out.
>
> **Strengths:** Thanks for your kind words!
>
> **Weakness:**
>
> 1. We believe that one of the main contributions of our work is identifying an assumption (i.e., (k,d)-adaptivity) under which a $d$-independent estimation is attainable. As a direct consequence of the assumption,  we are hence able to exploit the underlying martingale structure in terms like $X^\top_{ad}X_{nad}$ to obtain tighter bounds on them (please view our rebuttal to Reviewer Pp5b for more details), which contribute to the proof of Theorem 3.1 and 3.2. The main techniques used in the proofs are martingale concentration inequalities, matrix concentration inequalities, and matrix calculations (see e.g., Lemma A.1, A.2 for more details).  From our viewpoint, the proofs in Theorem 3.1, 3.2 are built based on basic concentration inequalities, and are intrinsically different from those in previous works, e.g., Theorem 8 in  Lattimore and Szepesvari, since our data assumption allows us to get tighter bounds on certain terms (e.g., $X^\top_{ad}X_{nad}$) in a different way which cannot be done without such an assumption.
>
> 2. We express our gratitude to the reviewer for their insightful question. Our immediate objective is to extend the TALE estimator to encompass multiple coordinates and incorporate $(k,d)$ adaptivity. To ensure asymptotic normality guarantees for all $k$-coordinates associated with $x_{ad}$, we need to construct $k$-dimensional weights $w$ which ensure asymptotic normality of the term $v_n$ and facilitate good control of the bias term $b_n$ (see equation (13) in the paper).  Currently, we are able to formulate weights $w$ that ensure the asymptotic normality of the term $v_n$. However, the conditions necessary for effectively controlling the bias term $b_n$, with these weights, are somewhat stringent based on our present calculations. In the revised version of our paper, we will provide an elaborate discussion on the methodology behind constructing these weights in $(k,d)$-settings, shedding light on the challenges posed in controlling the bias term. We extend our gratitude once again to the reviewer for their meticulous observations, which greatly contribute to the refinement of our work.
>
>
> 3. Thanks for the suggestion! We agree that the overuse of $\sigma$ might cause confusion in understanding the paper and we will fix it.
>
> **Questions:**  We use the term "adaptivity" to imply that the data are collected in an adaptive manner, in contrast to i.i.d. manner. This term is also used in other works [14, 21, 30], while another name "stochastic linear model" has also been used in history [23, 24]. We sincerely apologize if our use of the term "adaptivity" caused any unnecessary confusion.

---

> > ### Comment · Reviewer_KiDv · 2023-08-15
> > **Reply to the rebuttal**
> >
> > Dear authors,
> >
> > Thanks for your rebuttal. It is well received. I also read comments from other reviewers.
> >
> > 1. Since the bound (Eq. 7) from Theorem 3.1 is very similar to prior works (Eq. 8), it would be a good idea to discuss and highlight the technical differences in the revision, as emphasized in the rebuttal.
> > 2. Thanks for the detailed elaboration on challenges for extending to the more general cases. Discussing this point as a limitation of the present work would also be of interest to readers.
> >
> > I think the rebuttal adequately addressed my comments. I am happy to keep my positive evaluation of the paper.
> >
> > Regards, KiDv

---

> > > ### Author Response · Authors · 2023-08-17
> > >
> > > We greatly appreciate your thoughtful feedback and valuable recommendations. In the updated version of the paper, we intend to put additional emphasize the following key aspects:
> > >
> > > a) The significance of the (k,d) adaptivity, along with a comprehensive comparison of our analysis techniques in contrast to prior studies.
> > >
> > > b) We will incorporate our latest calculations pertaining to inference in the general (k,d) adaptive scenario, and will highlight the associated challenges.
> > >
> > > c) Additionally, we plan to integrate the simulated experiments that were included in the rebuttal phase.
> > >
> > > We sincerely hope that these enhancements will significantly elevate the paper’s quality.
> > > Please do let us know if you have any additional comments / suggestions.

---

### Official Review · Reviewer_tdeW · 2023-07-05

**Soundness:** 3 good
**Presentation:** 2 fair
**Contribution:** 3 good
**Rating:** 6
**Confidence:** 2

**Summary:**

In "Statistical Limits of Adaptive Linear Models: Low-Dimensional Estimation and Inference" the authors consider the problem of estimating a low-dimensional signal in a high-dimensional linear model where data collection is allowed to be adaptive. The notion of adaptivity employed in this paper restricts itself to k components, meaning that the k first (wlog) covariates are allowed to be adaptive while the remaining are assumed i.i.d. This is in contrast to prior work where all components are allowed to be adaptive which yields very detrimental minimax lower bounds even in the case where only one component is to be estimated. Based on this new notion of adaptivity, the authors propose a scheme to estimate the low-dimensional signal yielding beneficial scaling guarantees, both in mean square error and asymptotic normality.

**Strengths:**

1. Originality: the authors study a novel scenario and provide a novel scheme with reasonable guarantees. The work has sufficient novelty.
2. Quality: the paper is technically sound and the results are appropriately stated.
3. Clarity: The main results of the paper and some intuition on how they are established are clear. The example provided could be a bit clearer. I expand on this in the Questions section.
4. Significance: This may be to my own ignorance but the significance of the considered model is not entirely clear to me. I again will expand on this in the Questions section.

**Weaknesses:**

I merge this with the Questions section.

**Questions:**

1. Regarding Example 2.1.: Based on how the filtration is defined and the dimensionality of the quantities one can infer that A_i is considered to be the first coordinate of the covariates in the linear model. I don't see the loss of generality in instead defining x_i to include the treatment assignment as x_1i. This makes the mapping to the linear model more obvious.
2. Regarding significance of the model. The underlying assumption is that variable selection is not necessary in this case, i.e. the low-dimensional coordinates to be estimated are known prior, and they are the only ones that are affected by adaptive covariates while the remaining are i.i.d.. Can the authors expand on the treatment assignment example to provide with a situation in which this would be the case?

**Limitations:**

-

---

> ### Author Rebuttal · Authors · 2023-08-02
>
> We would like to thank the reviewer for taking the time to review our paper and provide helpful feedback! We truly appreciate your comments and suggestions, and believe they can make our work better.
>
> In the following we hope to address each of the points made in the review, in the order as they came out.
>
> **Summary and strengths:** Thank you for writing a detailed summary and for the kind words. We really appreciate them.
>
> **Questions:**
>
> 1. Thanks for the suggestion! Yes we implicitly think of the treatment $A_i$ as the first coordinate of the covariate vector. The reason we separated them is to follow the conventions in causal inference literature, in which the treatment and the covariate vector are denoted separately. Sorry for the confusion we made and thanks again!
>
> 2. First, we thank the reviewer for this engaging question. In the treatment assignment problem (Example 2.1), the goal is to estimate the treatment effect --- coordinate associated with $A_i$, and from the right-most bound in Corollary 3.3 we see that the estimation rate only depends on the degree of adaptivity $k$ (or an upper bound of it), but the location of the adaptive coordinate is unknown. Indeed, the centered OLS or the OLS do not use the location of adaptive coordinates in any way. In other words, in order to have an upper bound on the estimation error of the treatment assignment coordinate, we only need to have an upper bound on the number of regressors $k$ the treatment assignment $A_i$ depends on, and do not need to know the specific treatment allocation rule --- which gives you information regarding the location of these regressors. Finally, the rate in Corollary 3.3 is unimprovable due to our minimax lower bound from Proposition 2.2. We will clarify this important distinction in the modified version of the paper, and we thank the reviewer for pointing this out.

---

> > ### Comment · Reviewer_tdeW · 2023-08-14
> >
> > Thank you for your answer. I will keep my current score.

---

> > > ### Author Response · Authors · 2023-08-17
> > >
> > > Thanks for your insightful comments and helpful suggestions. Please do let us know if you have any additional comments / suggestions which might improve the quality of the paper.

---

### Official Review · Reviewer_Pp5b · 2023-07-06

**Soundness:** 3 good
**Presentation:** 3 good
**Contribution:** 3 good
**Rating:** 7
**Confidence:** 3

**Summary:**

This paper considers the issue of adaptive data collection in a linear regression model. To summarize the main idea, let us focus on the leading example in the paper (that is, Example 2.1, treatment assignment). In this example, a patient is treated based on effectiveness of the previous treatments as well as a small number of covariates. When the treatment is assigned adaptively in an unknown way, the treatment effect can be estimated via OLS only at a rate $\sqrt{d/n}$, where $d$ is the dimension of the entire covariate vector (not the dimension of the covariates used for treatment assignment) and $n$ is the sample size. This result is shown by previous work in Khamaru et al. [21] and a simplified version is given as Proposition 2.2 in the paper. One of the main results in the paper is that the centered OLS estimator of the treatment effect attains a better rate of convergence, that is, $\sqrt{k/n}$, where $k-1$ is the dimension of the covariates used for treatment assignment. The inference problem is further studied for the case that $k=1$. That is, the treatment assignment mechanism does not depend on the covariates but on the effectiveness of the previous treatments. The paper proposes an adaptive estimator called Two-stage Adaptive Linear Estimating Equation (TALE) Estimator. The adaptive weights are constructed in a particular fashion to develop asymptotic normality (see Theorem 3.4). Numerical experiments show potential usefulness of the proposed TALE estimator.

**Strengths:**

- This paper considers a highly important problem in the literature: adaptive data collection (e.g., bandits) is increasingly important in a number of fields.
- The paper clarifies the important open question in the literature, that is, "Can we obtain a good estimator for a low-dimensional parameter component in linear models when the degree of adaptivity is given?".
- The proposed TALE estimator has desirable theoretical properties and shows promising numerical results.

**Weaknesses:**

- The non-adaptive component $x_i^{\mathrm{nad}}$ is assumed to be independent of the adaptive component $x_i^{\mathrm{ad}}$ (see lines 84-85). This seems quite strong in the sense that if this is the case, we could just drop the non-adaptive component $x_i^{\mathrm{nad}}$ in the regression model and then we automatically obtain the $\sqrt{k/n}$ rate, provided that the variance of the new regression error, which now includes the omitted part $\theta^\top x_i^{\mathrm{nad}}$, is bounded by a constant that is independent of $d$. Using the scenario in Example 2.1 with $k=1$ (that is, the treatment assignment mechanism depends only on the effectiveness of the previous treatments), it might be preferable to consider the difference-in-means estimator (that is, to include only the intercept term and a treatment indicator) instead of estimating the treatment effect via regression adjustment. It would be useful to carefully discuss the issue of independence between the adaptive and nonadaptive components.

**Questions:**

- Line 191: it seems that the conditional variance $\sigma^2$ is a constant, meaning that it does not depend on $(x_i, \mathcal{F}_{i-1})$. This is a restrictive assumption and could be commented in line 198.
- The centered OLS is a proposed solution in the paper. I am wondering whether this estimator is the same as one that includes the intercept term. In other words, since it is conventional to use the intercept term in regression models, I am curious whether the standard practice already solves the research question raised in the paper (without fully realizing the importance of including the constant term in the regression model).
- The centered OLS algorithm on page 6 is not fully implementable in an online fashion. This is because computation of the sample means requires access to the full dataset. It might be useful to add some remarks regarding how to carry out online estimation for the centered OLS.
- Lots of notations are used before section 2.3. It might be better to move the notations section to improve readability of the paper.
- The TALE estimator is highly related to the concurrent work [2] entitled "Adaptive Linear Estimating Equations". It would be useful to clarify the differences between this work and the current paper.



**Limitations:**

The numerical results are promising but there is no theoretical result in the paper that implies that the TALE estimator should perform strictly better than W-decorrelation. It might be helpful to fully discuss what numerical results are predicted by asymptotic theory and what are not.

---

> ### Author Rebuttal · Authors · 2023-08-02
>
> We would like to thank the reviewer for taking the time to review our paper and provide helpful feedback! We truly appreciate your comments and suggestions, and believe they can make our work better.
>
> In the following we hope to address each of the points made in the review, in the order as they came out.
>
> **Summary and strengths:** Thank you for the detailed summary and positive feedback!
>
> **Weakness:**
>
> 1. Thanks for the question on the data assumption. We agree that the assumption might seem a bit strong, as the treatment is only allowed to be dependent on historical data and  $k-1$-th coordinates of the current covariate vector.  Regarding the alternative approach you proposed, we believe it is a good approach as long as the non-adaptive component $\theta^\top x^{nad}_i$ is small (independent of d) and has a near-zero mean. However, when the dimension of the non-adaptive component is relatively large, $\theta^\top x^{nad}_i$ may contribute significantly to the outcome $y_i$. Concretely, our assumption (A2) suggests that $\theta^\top x^{nad}_i$ is roughly of order $\nu | |\theta^*| |_2$, and therefore can not simply be bounded by a constant when $ | |\theta^*| |_2$ is large, e.g., $O(\sqrt{d})$ (note that we make no assumption on $ | |\theta^*| |_2$ in the paper).  Hope this help explain why we prefer OLS over the DID estimator.
>
> The requirement of the independence of $x_i^{ad}$ and $x_i^{nad}$ comes from a somewhat intrinsic technical difficulty. Assume $x_i^{nad}$ is zero mean for simplicity of discussion (otherwise we just consider $x_i^{nad}-E[x_i^{nad}]$).  Namely, in the analysis we are required to provide an $O(\sqrt{n})$ bound for $X_{ad}^\top X_{nad}=\sum_{i=1}^n x_i^{ad}x_i^{nad\top}$. To do so, we somehow inevitably need a martingale assumption on $x_i^{ad}x_i^{nad\top}$, and a sufficient condition for it to be a (matrix-valued) martingale is that $x_i^{nad}$ is independent of $x_i^{ad}$. (Note that we only need $x_i^{nad}$ to be zero-mean conditioned on $x_i^{ad}$, and the independence assumption used in Theorem 3.1 can be relaxed.)
>
>
>
> Technically speaking, the data assumption made in our work can be viewed as a complement to the quite strong assumption that the adaptive treatment assignment mechanism (i.e., the propensity score  $E[x_i^{ad}|x_{i}^{nad},\mathcal{F}_{i-1}]$)  is known [16, 30, 39, 41], since in the alternative case one can subtract the conditional expectation from $x_i^{ad}$, and therefore $(x_i^{ad}-E[x_i^{ad}])x_i^{nad\top}$ forms a martingale and enjoys an $O(\sqrt{n})$ bound.
>
> **Questions:**
>
> 1. Thanks for your advice. The equal conditional variance assumption is used in Thm 3.4 since it might be difficult to estimate the variance if it is time-dependent. The estimation results in Thm 3.1, 3.2 only require a uniform upper bound on the conditional variance. We have added a comment to line 198.
>
> 2. Yes from basic properties of linear models we know the centered OLS estimator is the same as the  (corresponding coordinates) of the OLS solution of a linear model with an additional intercept term. We choose to present it as the centered OLS estimator (which is also the standard practice) since it is more consistent with Thm 3.1 in which we assume the non-adaptive component is zero-mean and centering looks like a natural thing to do when the non-adaptive component has nonzero mean. Our main goal in Thm 3.1 and 3.2 was to show that one can have an improved single coordinate scaled-MSE than the pessimistic minimax lower bound under some reasonable data-collection assumptions, and we do not view the centered OLS as a novel estimator proposed.
>
>
> 3. As you already correctly pointed out that centered OLS is the same as OLS with an intercept. Consequently, one can use OLS with intercept formulation to obtain the centered OLS in an online fashion.
>
> 4. Thanks! We will fix this.
>
> 5. The focus of the paper "Adaptive linear estimating equations" is to establish a framework for inference for the whole parameter $\theta^\star \in R^d$ under $(d, d)$ adaptivity, leading to an estimate known as the ALEE estimate. However, it's worth noting that the ALEE estimate is not suitable for the high-dimensional regime, where the dimension $d$ is allowed to grow alongside the number of samples $n$. Additionally, the $1$-dimensional confidence intervals from the $d$ dimensional confidence regions produced by ALEE are too wide, especially in the case when $d$ is large.
>
> In our current work, our contribution is two-fold. First, we show that the estimation error for a single coordinate depends on the degree of dependence $k$ of that coordinate, and we can estimate that coordinate with a rate $\sqrt{k/n}$ -- much smaller than the minimax rate $\sqrt{d/n}$. Second,  we address the limitations of ALEE in high-dimensional models by introducing the Two-stage ALEE estimate (TALE). This novel approach is designed to be applicable and effective in the high-dimensional regime when there are additional structures present in the data, e.g. $(k, d)$ - adaptivity. Please also see the comments in the next paragraph.
>
>
> **Limitations:** Both TALE and W-decorrelation are designed to provide asymptotically normal inference, but they are focused on different regimes. The $W$-decorrelation estimator is designed for the fully adaptive regime ($(d,d)$ - adaptive regime), and may require the number of samples to be much larger than the dimension $d$ in worst cases. Indeed, in W-decorrelation type debiasing methods, the bias term decays at the rate $d / log (n)$ in the worst case, and these methods are only designed for the case when $d$ is fixed and the number of samples $n \gg e^d $.  In our current work, we are focused on the high dimensional case when the dimension of the problem $d$ can grow with the number of samples. Our main inference result (Thm 3.4) is that under $(1, d)$ adaptivity, we can perform valid one-dimensional inference when $d \approx \sqrt{n}$ up to some logarithmic terms.

---

> > ### Comment · Reviewer_Pp5b · 2023-08-18
> > **Thanks**
> >
> > I am grateful to the authors for their careful rebuttal. Most of my comments are well addressed. In view of that, I changed my rating upward by one point (from 6 to 7).

---

### Official Review · Reviewer_1ej5 · 2023-07-06

**Soundness:** 3 good
**Presentation:** 3 good
**Contribution:** 3 good
**Rating:** 5
**Confidence:** 3

**Summary:**

The paper introduces a new data collection assumption that captures the partially adaptive data and then derives a bound for scaled MSE of order $k\log n$, where $k$ is the number of entries that are collected adaptively.  Finally they also introduce a novel estimator for single coordinate inference which has  an asymptotic normality property.

**Strengths:**

The paper is clearly structured and well-written. The theorems seem solid, and the assumptions are general. The authors also provide a concrete example of $\textit{treatment assignment}$ to showcase the power of their theorems.

**Weaknesses:**

1. The numerical section is focused on the performance of TALE, missing out the numerical verifications for Theorem 3.1 and 3.2, which are the main theorems of the paper.

2. There still seem to be some fundamental limitations for the definition of $(k,d)$-adaptivity. For example, $(k,d)$-adaptivity requires a fixed number of entries in covariates $x$ are adaptively collected, which ignore the important scenario when such the number and indices of such entries may vary.

3. While the example of $\textit{treatment assignment}$  is very helpful, no comparison has been made against the state-of-the-art statistical tools and no numerical experiments are provided.

4. Typo in line 106.



**Questions:**

It is not very clear to me what is the technical difficulty to generalize the original bound involving $d$ (Lemma 16 of Lattimore and Szepesvari) to the bound of order $k$, and how the proof in this paper solves it.

**Limitations:**

See 2. in Weakness section.

---

> ### Author Rebuttal · Authors · 2023-08-02
>
> We would like to thank the reviewer for taking the time to review our paper and provide helpful feedback! We truly appreciate your comments and suggestions and believe they can make our work better.
>
> In the following, we hope to address each of the points made in the review, in the order they came out.
>
> **Strengths:** We are grateful for your positive feedback!
>
> **Weakness:**
>
> 1. Thanks for your valuable suggestion. Our main focus of the parameter estimation part (theorem 3.1, 3.2) is more theoretical-oriented, with the goal of understanding the theoretical limits (and the worst-case scenarios) of parameter estimation under different levels of adaptivity. Therefore we did not provide simulation results in the first version of our work. To address your concern,  we have conducted additional simulations regarding estimations by varying the levels of adaptivity.  More specifically, we keep $d$ fixed and varying the number of adaptive variables $k$. By adopting a data collection procedure similar to the construction in Kharmaru et al.[2], Lattimore [1],  we demonstrate the single coordinate estimation error grows linearly in $k$. See the global rebuttal and attached pdf for more details on the experiment.
>
> 2. Yes there are indeed some limitations in our definition of $(k,d)$-adaptivity. As the reviewer mentioned, the assumption does not directly apply to changing number of adaptive entries. One way to remedy this issue is to define $k$ to be the maximum number of adaptive entries across time. We agree that this solution is a bit crude when the number of adaptive coordinates varies a lot. One interesting and promising future direction is to understand the case when the number of adaptive coordinates is small for "most" of the samples.  We thank the reviewer for this engaging question.
>
> 3. Since throughout the paper we assume the data are generated from linear models, the state-of-the-art estimator people use is simply the OLS estimator, which is also semi-parametric efficient when data are i.i.d.  Other standard estimators for the average treatment effect (e.g., doubly robust estimators, IPW, AIPW) are more useful when the linear model assumption does not hold. In the current version of the paper, we focused on establishing the theoretical baseline for estimation and inference in the high dimensional adaptive linear model regime, and we chose simulated experiments for comparing various competing methods. We also conducted additional simulations which validate the theory developed in Theorems 3.1 and 3.2 of our paper, and we intend to include them in the final version of the paper. Our immediate future goal is to apply these methods to real datasets and see how these methods compare with existing state-of-the-art methods. We thank the reviewer for their thoughtful suggestion.
>
> 4. Fixed it. Thank you!
>
> **Question**: Thanks for asking the technical question. On one hand, Lemma 16 (or Theorem 8) of Lattimore and Szepesvari assumes completely adaptive covariates (i.e., $(d,d)$-adaptivity). Therefore, according to the lower bound in our Proposition 2.2,  the $d$ dependence in Theorem 8 of Lattimore and Szepesvari is unimprovable. On the other hand,  we can obtain $k$ (instead of $d$) dependence under $(k,d)-$adaptivity because the assumption implicitly imposes a martingale condition on the covariate vectors and hence allows us to have tighter bounds on terms like $X^\top_{\mathrm{ad}}X_{\mathrm{nad}}$. We refer the reviewer to Lemma A.3, A.4 and proofs of the theorems for more details.
>
> 1. Lattimore, Tor. "A Lower Bound for Linear and Kernel Regression with Adaptive Covariates." In The Thirty Sixth Annual Conference on Learning Theory, pp. 2095-2113. PMLR, 2023.
>
> 2. Khamaru, Koulik, Yash Deshpande, Tor Lattimore, Lester Mackey, and Martin J. Wainwright. "Near-optimal inference in adaptive linear regression." arXiv preprint arXiv:2107.02266 (2023).

---

> > ### Comment · Reviewer_1ej5 · 2023-08-15
> >
> > I thank the authors for carefully reading my review and providing detailed response. They adequately addressed my first point and gave a helpful discussion on my second and third points. However, I still agree with Reviewer KiDv that the technical contribution of the paper is incremental, so I will keep my rating unchanged.

---

> > > ### Author Response · Authors · 2023-08-17
> > >
> > > We are glad that the rebuttal was helpful, and extend our sincere gratitude for your insightful remarks. Please do let us know if you have any additional comments / suggestions which might improve the quality of the paper.

---

### Official Review · Reviewer_nCuW · 2023-07-10

**Soundness:** 3 good
**Presentation:** 3 good
**Contribution:** 3 good
**Rating:** 7
**Confidence:** 1

**Summary:**

As I have communicated with the area chair, I will not be reviewing due to a conflict of interest. Submitting default ratings intended to be ignored below.

**Strengths:**

NA

**Weaknesses:**

NA

**Questions:**

NA

---

### Official Review · Reviewer_4hmk · 2023-07-22

**Soundness:** 3 good
**Presentation:** 3 good
**Contribution:** 2 fair
**Rating:** 5
**Confidence:** 3

**Summary:**

This paper investigates degree of adaptivity in data impacts the performance of estimating a low-dimensional parameter component in high-dimensional linear models.  The main result is giving an error bound of a low-dimensional component that does not have diension dependence. They propose an estimator TALE. For a special case when there is a single adaptive coordinate and non-adaptive components have zero mean, this estimator is asymptotically normal.

The manuscript is easy to read through and the technical content of the paper appears to be correct albeit some typos:

Line 69: Adaptive
Line 478 in the appendix: $X_{\mathrm{ad}}$

**Strengths:**

Theoretical results:
- The authors define (k,d)-adaptivity to quantify the level of adaptivity along with a concrete example. Using this idea, they can get the upper bound depending on $k$ in stead of full dimension $d$. These results could bridge the gap between iid and arbitrarily adaptive data collection.

Experiments:
- Compared to other methods like OLS, estimation errors for TALE are in good accordance with a norm distribution.


**Weaknesses:**

Theoretical results:
- The main result in this paper demonstrates the advantages of utilizing the $(k,d)$-adaptivity structure, yielding a scaled-MSE bound in the order of $k\log(n)$ instead of $d\log(n). While this is a positive outcome, it's worth considering that the result is based on a different norm, which makes it less convincing.

Experiments:
- In the introduction,  the paper aims to obtain an estimator with performance dependent on the degree of adaptivity. Besides, the main results highlight that having $k$ in the upper bound. Therefore, I believe it would be better for the authors to experiment with different levels of adaptivity.


**Questions:**

Numerical experiments:
- The authors highlight that TALE exhibits shorter confidence intervals (CIs) compared to W-decorrelation, suggesting better estimation performance. However, it's noteworthy that OLS achieves much shorter CIs than TALE, especially in the higher-dimensional scenario where $d=50$. CI of TALE is about 50% longer than that of OLS. It would be helpful if the authors address this observation to provide a comprehensive evaluation of the methods. Also, given that, how could the authors conclude that TALE outperforms OLS in terms of estimation performance?
- From Figure 2, we can see that OLS is indeed downwardly biased. However, we can also observe that the magnitude of errors might be the same. The main results show a tighter bound (from $d=50$ to $k=1"), but the practical benefit is not evident. Additional analysis or insights would help demonstrate its significance.

**Limitations:**

In addition to the limitations mentioned earlier, the authors noted that they did not provide asymptotic normality guarantees when the adaptive component has more than one dimension.

---

> ### Author Rebuttal · Authors · 2023-08-02
>
> We would like to thank the reviewer for taking the time to review our paper and provide helpful feedback! We truly appreciate your comments and suggestions and believe they can make our work better.
>
> In the following we hope to address each of the points made in the review, in the order as they came out.
>
> **Typo: Line 69: Adaptive Line 478 in the appendix:**  Thanks for pointing out the typos! We have fixed them.
>
> **Strengths:**  Thanks for your kind words.
>
> **Weakness:**
> * First, we want to clarify that we use scaled-MSE as the metric in both our lower bound and upper bound results. Namely, eq. (5) and (7) are both bounds on the scaled-MSE when $\mathcal{I}=\mathrm{ad}$. Therefore we believe it is fair to conclude the improvement of the estimation error from $d\log(n)$ to $k\log(n)$ since they are computed under the same metric (see Proposition 2.2 and Theorem 3.1). We will clarify this in the final version of the paper. The reason we choose scaled-MSE as the metric is that we believe this is the most appropriate metric to use when evaluating an estimator in linear models.   As shown in eq. (1), under fixed design the scaled-MSE accounts for the variance $\mathbf{S}_n^{-1}$ induced by the design matrix, and is of order $O(1)$ for a single coordinate. Unlike $l_2$ norm or other more common metrics, the value of scaled-MSE is free from the choice of the designed matrix $X$ (while the $l_2$ error of one coordinate of  OLS is of order square root of the (1,1)-entry of  $\mathbf{S}_n^{-1}$), and thus is fairer to use when comparing the performance of estimators across different linear models.  In addition, we agree that in Theorem 3.2 (eq. 10) the metric we use does not exactly match the scaled-MSE, but they are the same whenever the non-adaptive component contains the all-one vector, which is often the case in practice.
>
> *  Thanks for the suggestion! Instead of proposing novel estimators for estimation, our main focus of the parameter estimation part is more theoretical-oriented, with the goal of understanding the theoretical limits (and the worst-case scenarios) of parameter estimation under different levels of adaptivity. Therefore, we did not include experiments on the level of adaptivity in the paper. However, to validate our theory, we conducted additional simulations by keeping $d$ fixed and varying $k$. By adopting a similar adaptive data collection procedure to the one presented in Khamaru et al.[20], we are able to show that the single coordinate estimation error grows linearly in $k$. Please see the global rebuttal and attached pdf for the simulations.
>
> **Question:** Thank the reviewer for pointing out the confusion in the experiment results. We hope our reply below can clarify the concerns.  As mentioned in the paper,  the reason we propose TALE is to obtain an estimator with inferential guarantee even when the data are adaptively collected. We completely agree that OLS usually has smaller estimation errors and smaller "confidence intervals" in practice than TALE. However, the issue of OLS is that it does not have valid inferential guarantees when data are adaptively collected, in contrast to the i.i.d. case. In Figure 1 (left), we see that OLS fails to have valid upper tail coverage (meaning the line corresponding to OLS is close to or above the baseline). In Figure 2 we see OLS estimator is not even unbiased. As a consequence, the "CI" deduced from OLS may not be valid, meaning that for example a 80% percent "CI" only covers the true parameter with 75% probability.
>
>  Therefore, the takeaway is that, if we only need an estimation of the true parameter without knowing how good or safe the estimation is, we should simply apply OLS. However, if we also want to know the quality of our estimation and provide confidence intervals for it, TALE might be a better option. In other words, TALE trades off accuracy for inferential guarantee, and we suggest people choose between OLS and TALE based on practical needs.
>
> In direct reply to your questions: 1). We are sorry that our writing might have left the impression TALE has a better estimation performance than OLS (line 306). As we mentioned we believe OLS likely has a better estimation performance, and what we did (line 306) was comparing the estimation performance between W-correlation and TALE. Sorry again for the confusion.     2). We agree that in Figure 2 the magnitude of errors of OLS and TALE seem to be the same.  This is actually in accordance with our theoretical results. As stated in Theorem 3.1, 3.4,  OLS gives O(1) scaled-MSE for single coordinate under (1,d)-adaptivity and zero-mean $X_{\mathrm{nad}}$,  which are assumed in the experiment setup, and TALE has near-optimal asymptotic variance. The main benefit of using TALE is its asymptotic normality guarantee instead of the estimation performance.  (Not sure if we understood your question correctly, feel free to let us know if you have more concerns.)
>
> **Limitations:**  We express our gratitude to the reviewer for their insightful question. Our immediate objective is to extend the TALE estimator to encompass multiple coordinates and incorporate $(k,d)$ adaptivity. To ensure asymptotic normality guarantees for all $k$-coordinates associated with $x_{ad}$, we need to construct $k$-dimensional weights $w$ which ensure asymptotic normality of the term $v_n$ and facilitate good control of the bias term $b_n$ (see equation (13) in the paper).  Currently, we are able to formulate weights $w$ that ensure the asymptotic normality of the term $v_n$. However, the conditions necessary for effectively controlling the bias term $b_n$, with these weights, are somewhat stringent based on our present calculations. In the revised version of our paper, we will highlight this limitation and provide an elaborate discussion on the methodology behind constructing these weights in $(k,d)$-settings, shedding light on the challenges posed in controlling the bias term.

---

> > ### Comment · Reviewer_4hmk · 2023-08-15
> >
> > I thank the authors for writing a detailed response to my questions and providing additional experiments. I'll raise my rating from 4 to 5.

---

> > > ### Author Response · Authors · 2023-08-17
> > >
> > > We are glad that the rebuttal was helpful, we sincerely thank you for your thoughtful comments. Please do let us know if you have any additional comments / suggestions which might improve the quality of the paper.

---

### Author Rebuttal · Authors · 2023-08-10

## Additional simulation:

As suggested by some of the reviewers, in this global rebuttal we display the effect of degree of adaptivity $k$ on the single coordinate estimation. These simulations provide empirical validation to the theory developed in the paper (Theorem 3.1, 3.2). To better visualize the result, we want to design an adaptive data collection, which can capture the estimation lower bound. Therefore, we adopt a similar data collection procedure as provided in Khamaru et al. [2] and also see Lattimore [1] for related information.

**Simulation set-up:**

1. Sample size $n = 1000$, $d = 300$.
2. The degree of adaptivity $k$ varies from 2 to 200 with step size equal to $3$.
3. Replication number $20$ for each level of adaptivity $(k,d)$.
4. $\theta^\mathrm{ad}_1 = 1$ and other coefficients are generated independently from $\mathcal{N}(0,1)$.
5. $\boldsymbol{x}_i^\mathrm{nad}$ is generated independently from uniform distribution on the sphere $\mathbf{S}^{d - k - 1}$ plus $\mathbf{E} \boldsymbol{x}_i^\mathrm{nad}$,
where  $\mathbf{E} \boldsymbol{x}_i^\mathrm{nad}$ is generated from $\mathcal{N}(0, \mathbf{I})$.

**Data collection method:**

Here we modified the data collection algorithm from Section $5.2.2$ in the paper [2]. The only difference between our data collection algorithm and the one in [2] is that we replace
$m_{u, v} \coloneqq \sum_{w=1}^v b_w\left(y_{u, w}-a_{u, w}\right)$ by
$$m_{u, v} \coloneqq \sum_{w=1}^v b_w \left(y_{u, w}-a_{u, w} - \boldsymbol{\theta}^{\mathrm{nad} \top} \boldsymbol{x}^\mathrm{nad}_{u+(w - 1)(d - 1)} \right).$$
In the modified data collection method above, we are able to address the non-adaptive components in the ridge estimates. See Section $5.2.2$ in paper [2] for more details.










**References:**
1. Lattimore, Tor. "A Lower Bound for Linear and Kernel Regression with Adaptive Covariates." In The Thirty Sixth Annual Conference on Learning Theory, pp. 2095-2113. PMLR, 2023.

2. Khamaru, Koulik, Yash Deshpande, Tor Lattimore, Lester Mackey, and Martin J. Wainwright. "Near-optimal inference in adaptive linear regression." arXiv preprint arXiv:2107.02266 (2023).

---

### Decision · Program_Chairs · 2023-09-21

**Decision:**

Accept (poster)

**Comment:**

This paper studies the effect of data adaptivity in the estimation of linear models. In particular, the authors investigate how the degree of adaptivity in data collection impacts the performance of estimating a low-dimensional parameter component in high-dimensional linear models. Toward this goal, the authors introduce the notion of (k,d)-adaptivity, which quantifies the amount of adaptivity in different coordinates of the linear model. In particular, a dataset is called (k,d)-adaptive if the first k coordinates of the covariates are designed adaptively, while the remaining d-k coordinates are iid. Under (k,d)-adaptivity, the authors show that the simple OLS estimator achieves a scaled-MSE error rate that scales with k. This is a generalization of the fully-adaptive scenario, in which the scaled-MSE scales with the dimension d, as well as a generalization of the fully non-adaptive case where the full dataset is assumed to be iid. For the special case of (1,d)-adaptive dataset, the authors provide a new algorithm, called TALE, which, unlike the simple OLS, enjoys an asymptotic normality property.

Five reviewers have reviewed the paper, and their overall assessment of the paper was positive. I agree with this assessment and believe the paper is a solid contribution with interesting results. Several reviewers have mentioned that the authors should highlight the key differences between their proposed proof technique and that of Khamaru et. al. [21]. I agree with this comment and encourage the authors to address this in their revised manuscript, as well as add the new complementary experiments to the main body of the paper.